# On the Expressiveness of State Space Models via Temporal Logics

**Eric Alsmann, Lowejatan Noori & Martin Lange**
Theoretical Computer Science / Formal Methods
University of Kassel, Germany
`{eric.alsmann,l.noori,martin.lange}@uni-kassel.de`

## Abstract

We investigate the expressive power of state space models (SSM), which have recently emerged as a potential alternative to transformer architectures in large language models. Building on recent work, we analyse SSM expressiveness through fragments and extensions of linear temporal logic over finite traces. Our results show that the expressive capabilities of SSM vary substantially depending on the underlying gating mechanism. We further distinguish between SSM operating over fixed-width arithmetic (quantised models), whose expressive power remains within regular languages, and SSM with unbounded precision, which can capture counting properties and non-regular languages. In addition, we provide a systematic comparison between these different SSM variants and known results on transformers, thereby clarifying how the two architectures relate in terms of expressive power.

## 1 Introduction

State Space Models (SSM) have emerged as a potential alternative to traditional sequence modelling architectures such as the popular transformer architecture. In this paper, we investigate the expressive capabilities of SSM by establishing lower bounds on their computational power. The analysis focuses on two key dimensions that influence expressiveness: the type of SSM layers used and the arithmetic precision employed in computations. We specifically examine diagonal-gated SSM like S6, cf. Gu & Dao (2024), where gate matrices can depend on the input but must maintain a diagonal structure, and time-invariant SSM like S4, cf. Gu et al. (2022), where gate matrices remain constant regardless of input. For each model variant, we consider both fixed-width arithmetic, with a constant number of bits for all computations, and log-precision arithmetic, where precision scales logarithmically with input length.

While empirical studies drive much of the progress in sequence modelling, foundational analysis provides complementary insights by establishing what architectures can and cannot represent in principle, independent of training dynamics. Motivated by the recent success of such analyses for transformers (Strobl et al. (2024)), we investigate SSMs using the same logical and complexity-theoretic framework. This allows for a precise *relative* comparison: we map SSM variants to the exact same logic fragments recently used to characterise transformer variants (e.g., UHAT, AHAT).

We structure our analysis around three conceptual levels of expressiveness. The most fundamental level is pattern matching (linear temporal logic / first order logic), which captures the ability to detect sequences of events based on their relative order without counting. Extending this, we consider modular predicates, which enable the tracking of periodic positions similar to what positional embeddings in transformer architectures do. Finally, we analyse global counting, which allows for comparing quantities over the entire sequence and strictly necessitates log-precision arithmetic.

By mapping SSMs to these levels, we provide structural guarantees on which patterns are theoretically learnable or provably impossible for a given architecture. We show that diagonal-gated SSM can recognise languages defined by pure-past Linear Temporal Logic on finite traces ($\mathrm{PLTL}_f$) with counting capabilities, while time-invariant SSM capture languages expressible in fragments with unary temporal operators plus modular predicates. Furthermore, we show that global counting strictly requires log-precision arithmetic. The lower bounds established in this paper contribute

to a growing body of research on the theoretical foundations of neural network architectures for sequence modelling, complementing empirical observations about their practical performance. Crucially, the resulting impossibility proofs identify fundamental architectural bottlenecks, such as the inability of diagonal fixed-precision SSMs to represent non-monotonic patterns like $(aa)^*$, which hold independently of the training data or optimization procedure. In this sense, our foundational characterisation complements empirical benchmarks by delineating which tasks are structurally out of reach for specific SSM variants.

RELATED WORK. Recent work on the theoretical foundations of SSM has been related to circuit complexity, cf. Merrill et al. (2024), automata theory, cf. Sarrof et al. (2024), and communication complexity, cf. Zubic et al. (2025). Alsmann & Lange (2025) investigated the computational complexity of verifying SSM. To the best of our knowledge, this is the first paper connecting the expressive power of SSM to formal logic.

The work on the formal expressiveness of SSM started with Merrill et al. (2024) who provided upper bounds via circuit complexity. They showed that diagonal gated as well as time-invariant SSM working over $\log$-precision as well as models working over fixed-precision (e.g. floating-point) are contained in $\text{TC}^0$. This is the class of functions definable with boolean circuits of polynomial size, constant depth and using threshold gates. They also showed that SSM with arbitrary gates, meaning that the gate can be an arbitrary input-dependent matrix, can recognise all regular languages including $\text{NC}^1$-complete languages. This implies a strict increase in expressiveness unless $\text{TC}^0 = \text{NC}^1$. Further work by Sarrof et al. (2024) provided a lower bound for diagonal gated SSM, proving that they can recognise all star-free languages. They also showed that under certain restrictions this lower bound is tight in the sense that these SSM recognise a regular language if and only if it is star-free. Our work significantly extends the lower bound provided by Sarrof et al. (2024). While they proved that diagonal SSMs (with a specific output function) can recognise star-free languages, our Theorem 1 recovers this as a special case within a broader hierarchy. Crucially, we extend the analysis along three new axes: (1) We characterise *time-invariant* and *mixed* SSMs, relating them to modular predicates ($\text{PLTL}_f[\text{MOD}]$); (2) we analyse the impact of arithmetic precision, showing that *log-precision* is strictly necessary for non-regular counting languages (like $a^n b^n c^n$); and (3) we integrate these findings into the existing transformer landscape (UHAT/AHAT), enabling a direct architectural comparison. Zubic et al. (2025) showed that SSM working over fixed-precision can only recognise regular languages.

Parallel to the work on the expressiveness of SSM there has been a lot of research on the expressiveness of transformer architectures via connections to logics (see Strobl et al. (2024) for a survey on these results). The most important work regarding this paper are Yang et al. (2024a), who proved connections between unique hard-attention transformers and first-order logic as well as Barcelo et al. (2024) and Yang et al. (2025), who investigate the counting abilities of several transformer architectures via extensions of linear temporal logic.

## 2 FUNDAMENTALS

STATE SPACE MODELS. We explore SSM in a generalised setting as presented in Merrill et al. (2024). For a structured analysis of this model, we will formalise its architecture following similar approaches as in Sarrof et al. (2024) and Merrill et al. (2024). An SSM layer $l$ is defined as a tuple $(\boldsymbol{h}_0, gate, inc, \phi)$, where $\boldsymbol{h}_0 \in \mathbb{R}^d$. The function $gate$ is a mapping $\mathbb{R}^d \to \mathbb{R}^{d \times d}$, $inc$ is a mapping $\mathbb{R}^d \to \mathbb{R}^d$, and $\phi$ is a mapping $\mathbb{R}^d \times \mathbb{R}^d \to \mathbb{R}^d$. An SSM layer transforms an input sequence of vectors $\boldsymbol{x}_1 \cdots \boldsymbol{x}_k \in (\mathbb{R}^d)^*$ into an output sequence $\boldsymbol{z}_1 \cdots \boldsymbol{z}_k \in (\mathbb{R}^d)^*$ through the following process: it computes an intermediate sequence $\boldsymbol{h}_1 \cdots \boldsymbol{h}_k \in (\mathbb{R}^d)^+$ via the linear recurrence

$$\boldsymbol{h}_t = gate(\boldsymbol{x}_t) \cdot \boldsymbol{h}_{t-1} + inc(\boldsymbol{x_t}) \quad \text{for } 1 \leq t \leq k$$

and subsequently generates the output via $\boldsymbol{z}_t = \phi(\boldsymbol{h}_t, \boldsymbol{x}_t)$.

An SSM $S$ comprising $L$ layers processes sequences over a finite alphabet $\Sigma$. It is expressed as a tuple $S = (emb, l_1, \cdots, l_L, out)$. Each $l_i$ refers to a layer as defined above, while $emb$ is a function $\Sigma \to \mathbb{R}^d$, and $out$ is a function $\mathbb{R}^d \to \mathbb{R}^d$. The SSM computes a function $\Sigma^* \to \mathbb{R}$ as follows: let $\sigma = \sigma_1 \cdots \sigma_k \in \Sigma^*$ be a word. Initially, the SSM computes the embedding $\boldsymbol{x}_1^0 \cdots \boldsymbol{x}_k^0$ of the word by setting $\boldsymbol{x}_i^0 = emb(\sigma_i)$. Subsequently, for each layer $1 \leq j \leq L$: compute

$\boldsymbol{z}_1^j \cdots \boldsymbol{z}_k^j = l_j(\boldsymbol{x}_1^{j-1} \cdots \boldsymbol{x}_k^{j-1})$. Each layer's output serves as the input for the succeeding layer, meaning $\boldsymbol{x}_1^{j+1} \cdots \boldsymbol{x}_k^{j+1} = \boldsymbol{z}_1^j \cdots \boldsymbol{z}_k^j$. The SSM's final output, denoted as $\boldsymbol{y}_1 \cdots \boldsymbol{y}_k$, is derived by applying $out$ element-wise: $\boldsymbol{y}_i = out(\boldsymbol{z}_i^L)$. In the end, the output of $S(\sigma)$ is computed by $\boldsymbol{y}_k$. We say that $S$ accepts a word $\sigma$ if $S(\sigma) = 1$. Otherwise it is rejected. We denote the language accepted by $S$ as $L(S)$.

Recent work on the expressiveness of SSM, cf. Merrill et al. (2024) and Sarrof et al. (2024), distinguishes between two main classes of SSM which differ in the allowed *gate* functions. The class of *time-invariant* SSM, cf. Mehta et al. (2023) and Orvieto et al. (2023), only allow layers with *gate* functions such that there is a matrix $A \in \mathbb{R}^{d \times d}$ with $gate(\boldsymbol{x}) = A$ for all $\boldsymbol{x} \in \mathbb{R}^d$. The class of *diagonal*-gated SSM, cf. Gu & Dao (2024), De et al. (2024) and Yang et al. (2024b), only allow layers that use *gate* functions such that $gate(\boldsymbol{x})$ is a diagonal matrix (with non-negative entries) but can depend on $\boldsymbol{x}$. Some SSM architectures like RetNet from Sun et al. (2023) use diagonal time-invariant gates. We call SSM which combine diagonal as well as time-invariant layers *mixed*. To the best of our knowledge, mixed SSM have not been used in practice yet, but they are interesting from a theoretical point of view as upper bounds on expressiveness for those directly apply to time-invariant *and* diagonal SSM. We therefore include them in our investigation. Lastly, we call SSM which do not have any restriction on the *gate*-mechanism, cf. Hasani et al. (2022), *arbitrary*. The global output $out$ and the output of each layer $\phi$ which also gets the initial input modelling a residual connection is computed by a non-linear function. In practice, different kinds of non-linear functions are used. In this paper we assume them to be represented by a Feedforward Neural Network (FNN) with ReLU activations.

ARITHMETICS. In our expressiveness analysis, we consider two distinct settings for the arithmetic precision of SSM computations. First, we examine fixed-precision arithmetic, where all values and computations use a constant number of bits $b$ (like floating-point or fixed-point representations), regardless of the input length. We say that an SSM works over fixed-precision arithmetic if all values and computations are carried out using only these $b$ bits. We additionally assume that the fixed-precision arithmetic is saturated, meaning that if an overflow occurs, the result is capped at the maximum representable value (resp. the minimum representable value for underflows) and that arithmetic operations are monotonic with respect to rounding. This is standard behaviour in, e.g., floating-point arithmetic, cf. 876 (2019). Second, we study log-precision arithmetic, where the precision grows logarithmically with the input length—specifically, using $O(\log n)$ bits for inputs of length $n$. This setting was studied with regards to expressiveness both for SSM by Merrill et al. (2024) and for transformer architectures by Merrill & Sabharwal (2023) and Hao et al. (2022), as it is both practically reasonable and theoretically significant: it provides sufficient precision to accurately count occurrences or compute sums that grow linearly with input size, while avoiding the unrealistic assumption of unbounded precision. This balanced approach is particularly interesting because it offers a middle ground between the restrictive fixed-precision model and the impractical unbounded precision model.

LINEAR TEMPORAL LOGIC OVER FINITE TRACES. We use linear temporal logic over finite traces (LTL$_f$) as introduced by De Giacomo & Vardi (2013) to analyse the expressiveness of SSM. It extends propositional logic with temporal operators, enabling the expression of properties that involve the ordering and timing of events. We consider the pure-past fragment PLTL$_f$ of LTL$_f$ as studied by De Giacomo et al. (2021), which uses the temporal operators *yesterday* Y ("at the previous position"), *previously* P ("at some point in the past") and *since* S ("since some event occured, another event occured continuously"). Let $\mathcal{P}$ be a finite set of atomic propositions. The syntax of PLTL$_f$ is defined as follows:

$$\varphi ::= p \mid \neg\varphi \mid \varphi \wedge \varphi \mid \mathsf{Y}\,\varphi \mid \mathsf{P}\,\varphi \mid \varphi\,\mathsf{S}\,\varphi$$

Formulas of $\mathrm{PLTL}_f$ are interpreted over finite words over the alphabet $\Sigma = 2^{\mathcal{P}}$. Given a word $\sigma = \sigma_1 \cdots \sigma_n \in \Sigma^*$ and $i \in \{1, \ldots, n\}$, the semantics of $\mathrm{PLTL}_f$ is inductively defined as follows:

$$
\begin{aligned}
\sigma, i &\models p & \iff & \quad p \in \sigma_i \\
\sigma, i &\models \neg\varphi & \iff & \quad \sigma, i \not\models \varphi \\
\sigma, i &\models \varphi \wedge \psi & \iff & \quad \sigma, i \models \varphi \text{ and } \sigma, i \models \psi \\
\sigma, i &\models \mathtt{Y}\,\varphi & \iff & \quad i > 1 \text{ and } \sigma, i-1 \models \varphi \\
\sigma, i &\models \mathtt{P}\,\varphi & \iff & \quad \exists\, 1 \le k \le i : \sigma, k \models \varphi \\
\sigma, i &\models \varphi\,\mathtt{S}\,\psi & \iff & \quad \exists\, 1 \le k \le i : \sigma, k \models \psi \text{ and f.a } k < j \le i : \sigma, j \models \varphi
\end{aligned}
$$

We say that $\sigma$ is a model of $\varphi$ iff $\sigma, n \models \varphi$ (or simply $\sigma \models \varphi$). Furthermore, we denote the language of a formula as $L(\varphi) = \{\sigma \in \Sigma^* \mid \sigma \models \varphi\}$. We use typical abbreviations $\mathtt{tt}$ (always evaluates to true), $\mathtt{ff}$ (always evaluates to false) and $\mathtt{H}\,\varphi = \neg\,\mathtt{P}\,\neg\varphi$ ($\varphi$ holds on all past positions). Furthermore, we call the unary fragment of $\mathrm{PLTL}_f$, which only uses *yesterday* and *previously*, $\textsc{un-}\mathrm{PLTL}_f$.

COUNTING EXTENSIONS. Yang & Chiang (2024) and Barcelo et al. (2024) analysed the expressive power of several transformer architectures, regarding the ability to do counting, by extending $\mathrm{LTL}_f$ with a counting operator. $\mathrm{PLTL}_f[\overleftarrow{\#}, \overrightarrow{\#}]$ extends $\mathrm{PLTL}_f$ syntactically by one additional case: if $\varphi_1, \cdots, \varphi_i, \psi_1, \cdots, \psi_j$ are $\mathrm{PLTL}_f[\overleftarrow{\#}, \overrightarrow{\#}]$ formulas, $a_1 \cdots a_i, b_1, \cdots b_j \in \mathbb{Z}$, $c \in \mathbb{N}$ and $\sim \in \{<, \le, =, \ge, >\}$ then

$$
\sum_i a_i \overleftarrow{\#}\,\varphi_i + \sum_j b_j \overrightarrow{\#}\,\psi_j \sim c
$$

is also a $\mathrm{PLTL}_f[\overleftarrow{\#}, \overrightarrow{\#}]$ formula. When evaluating a formula on a word $\sigma$ and position $i$, $\overleftarrow{\#}\,\varphi$ counts the number of positions $j \le i$ such that $\sigma, j \models \varphi$, and $\overrightarrow{\#}\,\varphi$ does so analogously for positions $j \ge i$.

By $\mathrm{PLTL}_f[\overleftarrow{\#}]$ we denote the logic in which every counting subformula has $b_j = 0$ for all $j$. The use of $\overleftarrow{\#}$ (and not $\overrightarrow{\#}$) is tied to the nature of SSMs: unlike transformers with full sequence attention, SSMs only have access to the prefix of the input at each step.

MODULAR PREDICATES. We introduce a third variant of $\mathrm{PLTL}_f$ which adds *modular predicates*. The syntax of $\mathrm{PLTL}_f[\mathtt{MOD}]$ is extended by additional atomic formulas $\mathtt{MOD}_r^m$ for $m > 1$ and $0 \le r < m$. The semantics is the same as in $\mathrm{PLTL}_f$. For the new operator it is extended by

$$
\sigma, i \models \mathtt{MOD}_r^m \iff i \equiv r \pmod{m} .
$$

Crucially, allowing these modular predicates strictly increases the expressive power beyond star-free languages (which correspond to standard $\mathrm{PLTL}_f$). It enables the definition of periodic properties, such as recognising the language $(aa)^*$, which necessitates distinguishing even from odd positions. Formally, this extension lifts the expressiveness to the class of all regular languages definable in $\mathrm{AC}^0$ (Straubing, 1994). This complexity class serves as a key reference point in our later analysis, as we will show that it aligns with the capabilities of mixed SSMs and transformers equipped with positional encodings.

## 3 RESULTS

Our analysis establishes lower bounds on SSM expressiveness across different architectural variants and precision settings. The results reveal an interesting hierarchy where expressiveness varies significantly based on gating mechanisms and arithmetic precision. An overview of these findings is illustrated in Figure 1. It also shows how our fragments of $\mathrm{PLTL}_f$ relate to circuit complexity and first-order logic. A detailed explanation of these connections can be found in Appendix A.1.

DIAGONAL SSM. Theorem 1 demonstrates that diagonal SSM with fixed-width arithmetic can recognise the languages definable in $\mathrm{PLTL}_f$, which corresponds to the class of first-order definable or star-free regular languages. This result is constructive: we show how to systematically translate any $\mathrm{PLTL}_f$ formula into an equivalent diagonal SSM by decomposing the formula according to its nesting depth and implementing each temporal operator through appropriate gate mechanisms.

The key insight underlying this construction is that while temporal operators like Y and P can be implemented using simple gating patterns, the *since*-operator requires input-dependent diagonal gates due to its recursive definition $\varphi \,\text{S}\, \psi \equiv \psi \vee (\varphi \wedge \text{Y}(\varphi \,\text{S}\, \psi))$. This dependency on current input evaluation necessitates the diagonal structure of the gates.

We complement this lower bound with a matching non-expressibility result. Theorem 3 proves that diagonal SSM with fixed precision cannot recognise the simple non-star-free language $(aa)^*$. This limitation arises from a fundamental monotonicity property (Lemma 2): when a diagonal SSM repeatedly processes the same input symbol, its output must eventually stabilise.

When diagonal SSM are equipped with logarithmic precision arithmetic, their expressiveness expands considerably. Theorem 4 shows that these models can recognise all languages definable in $\text{PLTL}_f[\overleftarrow{\#}]$, pure-past LTL extended with backward-looking counting operators. Barcelo et al. (2024); Yang & Chiang (2024) also used this logic to investigate the expressiveness of transformer architectures. The extension by counting formulas enables the recognition of non-regular (and even non context-free) languages, such as $\{a^n b^n c^n \mid n \geq 0\}$, which can be expressed by

$$\varphi = \text{H}\left((a \to \neg\,\text{P}(b \vee c)) \wedge (b \to \neg\,\text{P}\,c)\right) \wedge (\overleftarrow{\#}\,a - \overleftarrow{\#}\,b = 0) \wedge (\overleftarrow{\#}\,c - \overleftarrow{\#}\,b = 0)$$

TIME-INVARIANT SSM.   While we conjecture that time-invariant SSM cannot express the temporal dependencies captured by the since-operator (Conjecture 1), they possess another capability: the computation of modular predicates about sequence positions. Lemma 6 demonstrates how time-invariant SSM can maintain counters modulo $m$ using cyclic permutation matrices, enabling recognition of languages like $(aa)^*$ that are beyond the reach of diagonal SSM with fixed precision.

This leads to the characterisation in Theorem 7: time-invariant SSM with fixed precision recognise all languages definable in $\text{UN-PLTL}_f[\text{MOD}]$, the unary fragment of pure-past LTL extended with modular predicates. With logarithmic precision, they can additionally handle counting operators, recognising languages in $\text{UN-PLTL}_f[\text{MOD}, \overleftarrow{\#}]$.

MIXED AND ARBITRARY GATES.   Our hierarchy is completed by considering SSM that combine multiple gating mechanisms. Corollary 8 establishes that mixed SSM (combining both diagonal and time-invariant layers) with fixed precision can recognise all regular languages in $\text{AC}^0$, effectively capturing the union of capabilities from both individual architectures. SSM with arbitrary gates achieve even greater expressiveness, recognising all regular languages as established in prior work by Merrill et al. (2024).

## 4   EXPRESSIVE POWER OF DIAGONAL SSM

We show that diagonal SSM are at least as expressive as $\text{PLTL}_f$, which is expressively equivalent to the set of star-free languages. This follows the lines of Sarrof et al. (2024) but we use it as the base for further constructions regarding extensions and restrictions of $\text{PLTL}_f$.

In order to evaluate a $\text{PLTL}_f$ formula $\varphi$ at position $i$ of a word $\sigma$, one can evaluate the subformulas of $\varphi$ in a bottom-up manner. We order the subformulas in such a way that every subformula is evaluated after its own subformulas. Independent subformulas can be evaluated in parallel (in a single SSM layer). This means that the number of layers needed for all our constructions corresponds to the depth of the syntax tree of formula $\varphi$, here called *nesting depth* for brevity.

**Definition 1.** The *nesting-depth* $\text{nd}(\varphi)$ of a $\text{PLTL}_f$ formula $\varphi$ is defined inductively: $\text{nd}(p) = \text{nd}(\text{MOD}_r^m) = 0$, $\text{nd}(\varphi \circ_{\text{bin}} \psi) = \max(\text{nd}(\varphi), \text{nd}(\psi)) + 1$, $\text{nd}(\circ_{\text{un}}\varphi) = \text{nd}(\varphi) + 1$ and $\text{nd}\left(\sum_{j=1}^{k} a_j \cdot \overleftarrow{\#}\,\varphi_j \sim c\right) = \max\{\text{nd}(\varphi_j) \mid 1 \leq j \leq k\} + 1$ with $\circ_{\text{bin}} \in \{\wedge, \text{S}\}$ and $\circ_{\text{un}} = \{\neg, \text{P}, \text{Y}\}$.

This allows us to order the subformulas of $\varphi$ in the way described above. For each subformula $\psi$ of $\varphi$, we construct a diagonal SSM layer $l_\psi$ which computes for each position $i$ whether $\sigma, i \models \psi$. The SSM $S_\varphi$ for the whole formula $\varphi$ then computes whether $\sigma, n \models \varphi$. A similar idea has already been used by Alsmann & Lange (2025) to show that the satisfiability problem for diagonal SSM over fixed-precision is PSPACE-complete.

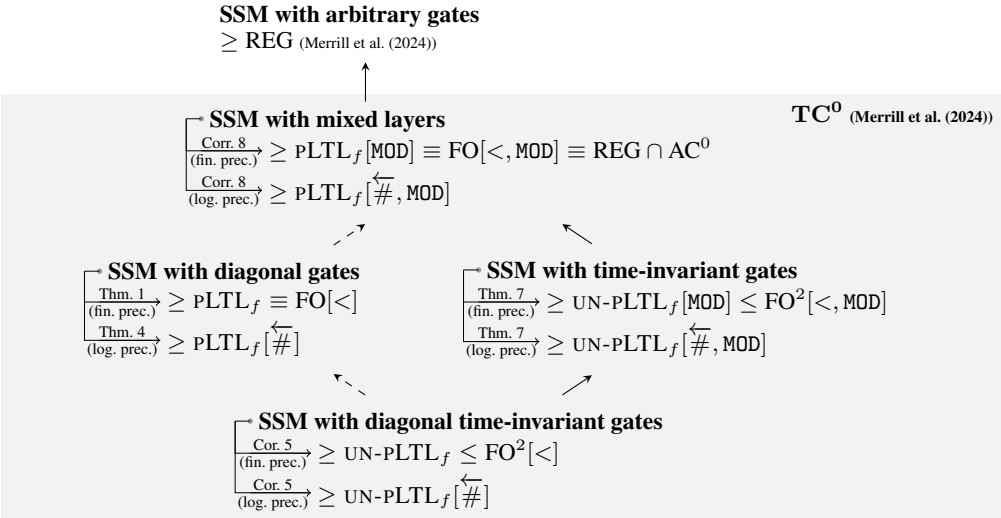

Figure 1: Expressiveness hierarchy of SSM architectures mapped to logical fragments and complexity classes. We establish lower bounds for Diagonal, Time-Invariant, and Mixed SSMs, distinguishing between fixed-precision and log-precision (enabling counting operators $\overleftarrow{\#}$). Dashed arrows indicate provably strict inclusions (Theorem 3). The gray box delineates the $TC^0$ upper bound from prior work (Merrill et al., 2024).

**Theorem 1.** *Diagonal SSM with fixed-precision recognise all languages definable in* $\text{PLTL}_f$.

*Proof sketch.* The detailed proof can be found in Appendix A.2. Each subformula corresponds to one dimension of the SSM's hidden states. The increment function of each layer together with the position-wise FNN add information about subformulas evaluated in previous layers and take care of the non-temporal subformulas such as conjunctions, negations and comparisons. After each layer, each hidden state is a boolean vector, indicating which subformulas are satisfied at each position. The temporal operators Y and P can be implemented by diagonal and time-invariant gates, because they do not require any conjunctions with the current input. Y consistently references the immediately preceding position, and P accumulates occurrences up to the current point. In contrast, the *since*-operator is inherently recursive, defined by $\varphi \mathsf{S} \psi \equiv \psi \vee (\varphi \wedge \mathsf{Y}(\varphi \mathsf{S} \psi))$, thus combining previous and current states through conjunction. Evaluating this operator inherently depends on the current input evaluation of $\varphi$, necessitating an input-dependent gate. $\square$

Next, we show that this characterisation is tight in the sense that diagonal SSM working over fixed-precision cannot recognise (aa)*, which is not definable in $\text{PLTL}_f$. This non-expressiveness result is based on a monotonicity property of diagonal SSM working over fixed-precision. Any diagonal SSM working over fixed-precision, after repeatedly seeing that same input must eventually become constant.

**Lemma 2.** *Let* $\mathcal{S}$ *be a diagonal SSM over an alphabet* $\Sigma$. *For* $\sigma \in \Sigma$, *let* $f_\sigma : \mathbb{N} \to \mathbb{F}^d$ *be the function defined by* $f_\sigma(n) = \boldsymbol{y}_n$, *where* $\boldsymbol{y}_n$ *is the last vector of the output sequence of* $\mathcal{S}$ *after the last layer on the word* $\sigma^n$. *Then there exists a number* $N \in \mathbb{N}$ *such that for all* $n \geq N$: $f_\sigma(n) = f_\sigma(N)$.

*Proof.* The key insight is that since the gate matrices are diagonal, each dimension of the hidden state evolves independently. For the first layer this means that for each dimension $i$, $(\boldsymbol{h}_t)_i = gate(emb(\sigma))_i \cdot (\boldsymbol{h}_{t-1})_i + inc(emb(\sigma))_i$. Because the *gate* is non-negative, this sequence must be monotonic. Given the fixed-precision arithmetic, each dimension can only take on a finite number of distinct values. Therefore, as $t$ increases, each dimension must eventually stabilize to a constant value. This implies that there exists some $N$ such that for all $n \geq N$, the output of the first layer remains constant. The same argument applies inductively to each subsequent layer, leading to the conclusion that the entire SSM's output stabilises after a finite number of repetitions of the input symbol. $\square$

The following is a direct consequence of Lemma 2 and the fact that $(aa)^*$ is not monotonic in the sense that there are words in the language that can be extended to a word not belonging to the language and vice-versa.

**Theorem 3.** *No diagonal SSM with fixed-precision can recognise $(aa)^*$.*

*Proof.* Assume that there exists a diagonal SSM $\mathcal{S}$ recognising $(aa)^*$. By Lemma 2, there exists a number $N$ such that for all $n \geq N$: $f_a(n) = f_a(N)$. This means that $\mathcal{S}$ either accepts or rejects all words of the form $a^n$ with $n \geq N$, contradicting the assumption. $\square$

Lemma 2 showed that diagonal SSM behave montonically for each input symbol. With diagonal SSM, even though they have a monotonic behaviour for each input symbol, they can still "reset" the hidden state when seeing a different symbols and thus evaluate the *since*-operator.

Having characterised the expressiveness of diagonal SSM working over fixed-precision, we now turn to diagonal SSM working over log-precision. In this setting, diagonal SSM can also recognise non-regular and even non-context-free languages. This was already observed by Sarrof et al. (2024). Also, Alsmann & Lange (2025) showed that the satisfiability problem for this class of SSM is undecidable. We show that diagonal SSM working over log-precision can recognise all languages definable in $\text{PLTL}_f$ extended by the *backward-looking* counting operator $\overleftarrow{\#}$.

**Theorem 4.** *Diagonal SSM with log-precision recognise all languages definable in $\text{PLTL}_f[\overleftarrow{\#}]$.*

*Proof sketch.* The detailed proof can be found in Appendix A.2. The proof is an extension of the proof of Theorem 1. Given a counting subformula $\sum_{j=1}^{k} a_j \cdot \overleftarrow{\#} \varphi_j \sim c$, we add one layer to the SSM which counts the occurrences of each $\varphi_j$ in the hidden state using the increment function. The position-wise FNN then checks whether their linear combination satisfies the comparison with $c$ and writes the result into the corresponding dimension of the hidden state. The rest of the construction remains unchanged. As this construction does not require any non-diagonal gates, it also works for time-invariant diagonal SSM. $\square$

## 5 EXPRESSIVE POWER OF TIME-INVARIANT SSM

As established in the previous section, diagonal SSM can evaluate all formulas in $\text{PLTL}_f$. However, the *since*-operator inherently requires input-dependent gates. This raises the question of whether time-invariant SSM, which have constant gates, can evaluate the *since*-operator. We conjecture that they cannot, and thus cannot recognise all languages definable in $\text{PLTL}_f$.

**Conjecture 1.** *No time-invariant SSM with fixed-precision can recognise $L(a\text{S}b)$ over $\Sigma = \{a, b, c\}$.*

The intuition behind this conjecture is that the *since*-operator requires combining information from the current input with information from previous positions in a way that depends on the current input. While time-invariant SSM can "reset" the hidden state when seeing a different symbol, time-invariant cannot adapt their *gate* based on the current input symbol. Due to this limitation, the only difference between seeing a $b$ or $c$ in a word, lies within the additive *inc*-part of the SSM. Our suspicion is that time-invariant SSM trying to recognise $a \text{ S } b$ will eventually reach a point, where they loose track of whether the last non-$a$ symbol was a $b$ or a $c$. Good candidates for words where this happens are $\sigma = a^n(ca^nba^n)^m$ and $\sigma' = a^n(ba^nca^n)^m$ for large $n, m$. A formal proof of this conjecture however is difficult for two reasons. First, the behaviour of time-invariant SSM is more complex than that of diagonal SSM, as the dimensions of the hidden state can interact with each other. Second, due to saturation effects in fixed-precision arithmetic and the fact the fixed-precision arithmetic is not associative, the behaviour of time-invariant SSM is difficult to analyse.

As the *since*-operator is the only temporal operator which needs an input-dependent diagonal gate we get the following immediately for the case in which the SSM layers are both diagonal and time-invariant. Without the *since*-operator, diagonal and time-invariant SSM can recognise languages in the unary fragment of $\text{PLTL}_f$.

**Corollary 5.** *SSM with layers that are both diagonal and time-invariant can recognise all languages definable in $\text{UN-PLTL}_f[\overleftarrow{\#}]$ under log-precision arithmetic, and those definable in $\text{UN-PLTL}_f$ under fixed-width arithmetic.*

While being possibly less expressive than diagonal SSM, time-invariant and non-diagonal SSM can still recognise languages which are not definable in $\text{PLTL}_f$. We demonstrate this by showing that time-invariant SSM can compute modular predicates about sequence positions. For any position $t$ in a sequence, modular predicates determine whether $t$ has a specific remainder when divided by some modulus $m$. This ability originates from the possibility of time-invariant SSM to maintain a counter modulo $m$ in their hidden state using a cyclic permutation matrix as the gate.

**Lemma 6.** *For any integer $m \geq 2$, there exists a time-invariant SSM layer $l_{\text{MOD}^m}$ that outputs $\boldsymbol{e}_r \in \mathbb{R}^d$ at position $t$ if and only if $t \equiv r \pmod{m}$.*

*Proof.* We construct the SSM layer $l_{\text{MOD}^m}$ as follows. The initial state is $\boldsymbol{h}_0 = \boldsymbol{e}_1$, the first standard basis vector. The gate matrix $P$ is the cyclic permutation matrix that maps $\boldsymbol{e}_i$ to $\boldsymbol{e}_{i+1}$ for $i = 1, \ldots, m-1$ and $\boldsymbol{e}_m$ to $\boldsymbol{e}_1$. Specifically, all of $P$'s entries are zero except for ones on the subdiagonal and in position $(1, m)$:

$$P = \begin{pmatrix} 0 & 0 & 0 & \cdots & 0 & 1 \\ 1 & 0 & 0 & \cdots & 0 & 0 \\ 0 & 1 & 0 & \cdots & 0 & 0 \\ \vdots & \vdots & \ddots & \ddots & \vdots & \vdots \\ 0 & 0 & 0 & \cdots & 1 & 0 \end{pmatrix}.$$

The increment matrix $B$ is set to zero, since the predicate depends only on the position and not on the input. The key insight is that the hidden state evolves as $\boldsymbol{h}_t = P^t \boldsymbol{h}_0 = P^t \boldsymbol{e}_1$. Since $P$ is a cyclic permutation of order $m$, we have $P^m = I$, and thus $P^t \boldsymbol{e}_1 = \boldsymbol{e}_{(t \bmod m)+1}$. This means that, at each position $t$, the hidden state is exactly the standard basis vector whose index encodes the remainder of $t$ modulo $m$. At position $t$, the state $\boldsymbol{h}_t$ has a 1 in position $(t \bmod m) + 1$ and 0's elsewhere. $\square$

**Example 1.** To illustrate the concept of modular predicates, consider $\text{MOD}_1^2$, which determines whether a position is odd. Here $m = 2$, so the permutation matrix is $P = \begin{pmatrix} 0 & 1 \\ 1 & 0 \end{pmatrix}$ and the initial state is $\boldsymbol{h}_0 = (1,0)^T$. The state sequence alternates between $\boldsymbol{e}_1 = (0,1)^T$ at odd positions and $\boldsymbol{e}_0 = (1,0)^T$ at even positions.

The cyclic permutation effectively maintains a counter modulo $m$ in the hidden state, enabling the computation of any modular predicate with a single time-invariant SSM layer. This especially allows non-star-free languages to be defined, e.g. $L(\text{H}\, a \wedge \text{MOD}_0^2) = (aa)^*$, which are not definable in $\text{PLTL}_f$, cf. Straubing (1994). In this sense diagonal and time-invariant SSM are incomparable in expressiveness, as diagonal SSM cannot recognise $(aa)^*$ and time-invariant SSM seem to be unable to recognise $L(a\, \text{S}\, b)$.

**Theorem 7.** *Time-invariant SSM with fixed-precision recognise all languages definable in $\text{UN-PLTL}_f[\text{MOD}]$ and time-invariant SSM with $\log$-precision recognise all languages definable in $\text{UN-PLTL}_f[\text{MOD}, \overleftarrow{\#}]$.*

*Proof sketch.* The detailed proof can be found in Appendix A.3. It is similar to the proof of Theorem 1. The main difference is that we need to add one layer for each modular predicate $\text{MOD}_r^m$ appearing in the formula. This layer is constructed as described in Lemma 6. The rest of the construction remains unchanged. As this construction does not require any non-diagonal gates, it also works for time-invariant diagonal SSM. $\square$

Having established lower bounds for diagonal and for time-invariant SSM, it seems natural to consider SSM that are either diagonal or time-invariant in each layer. Mixed SSM can evaluate both the *since*-operator and modular predicates. This allows us to show that mixed SSM with fixed-precision can recognise all languages in $\text{PLTL}_f[\text{MOD}]$ which are exactly the regular languages in $\text{AC}^0$, cf. Straubing (1994), the class of languages recognised by constant-depth polynomial-size circuits with unbounded fan-in. This class contains all star-free languages as well as languages like $(aa)^*$. The same holds for mixed SSM with $\log$-precision and $\text{PLTL}_f[\text{MOD}, \overleftarrow{\#}]$.

**Corollary 8.** *Mixed SSM with fixed-precision recognise all regular languages in $\text{AC}^0$. With $\log$-precision they recognise all regular languages in $\text{PLTL}_f[\overleftarrow{\#}, \text{MOD}]$.*

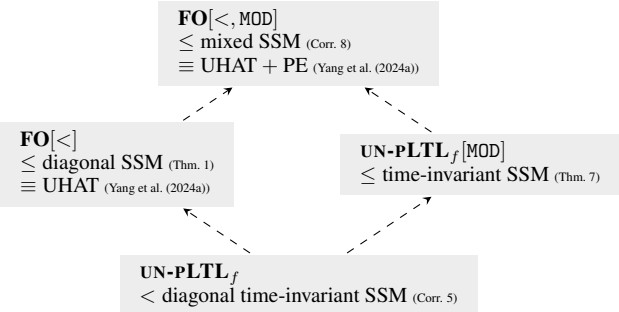

Figure 2: Detailed comparison of fixed-precision SSM variants with Unique Hard-Attention Transformers (UHAT) Yang et al. (2024a). The diagram illustrates a structural alignment: Diagonal SSMs capture star-free languages (FO[$<$]), matching the capabilities of UHAT without positional encodings. Time-invariant layers introduce modular predicates, providing an expressive lift analogous to adding Positional Encodings to transformers. Consequently, mixed SSMs capture the full class FO[$<$, MOD] (regular languages in $AC^0$).

## 6 COMPARISON TO TRANSFORMER ARCHITECTURES

In this section we will discuss how the results on the expressiveness of SSM fit into the broader landscape of recent results on SSM expressiveness and the comparative expressivity of various transformer architectures.

Transformer architectures studied in the formal expressiveness communities can be classified into two main categories: those with hard attention mechanisms and those with soft attention mechanisms. Hard attention transformers, e.g. unique hard-attention (UHAT) and average hard-attention (AHAT), use a discrete attention mechanism that allows them to focus on specific parts of the input sequence. In contrast, soft attention transformers (SAT), like the original transformer model by Vaswani et al. (2017) and its variants, employ a continuous attention mechanism that computes weighted averages over the entire input sequence. While unique hard-attention transformers can only attend to one position in the sequence, average hard-attention and soft-attention transformers can attend to multiple positions, allowing them to count occurrences of certain properties. This difference corresponds to the difference between fixed-precision and log-precision arithmetic in SSM.

Additionally due to the nature of the attention mechanism, transformers cannot distinguish between different positions in a sequence without additional positional encodings. These encodings provide the model with information about the order and position of elements in the sequence. In contrast, SSM inherently encode positional information through their recurrent structure. But as observed in Section 5, allowing time-invariant SSM layers increases expressivity because the model can also maintain a counter modulo $m$ in its hidden state, enabling them to compute modular predicates about sequence positions. Just as allowing time-invariant SSM layers, adding positional encodings to transformers also allows them to compute modular predicates, cf. Yang et al. (2024a) and Barcelo et al. (2024).

Yang et al. (2024a) established that unique hard-attention transformers (UHAT) can recognise all languages definable in first-order logic with the order predicate (FO[$<$]) and that adding positional encodings (UHAT+PE) allows them to recognise all languages definable in FO[$<$, MOD]. This aligns with our results on diagonal and time-invariant SSM working on fixed-precision, as can be seen in Figure 2. Analogously, Barcelo et al. (2024) showed that average hard-attention transformers (AHAT) can recognise all languages definable in $\text{PLTL}_f[\overleftarrow{\#}, \overrightarrow{\#}, \text{MOD}]$. This aligns with our results on mixed SSM working on log-precision. The increase in power of AHAT is due to the transformers' ability to attend to all positions in the input sequence. Yang & Chiang (2024) investigated the counting abilities of soft-attention transformers (SAT) by analysing a logic called C-RASP and its extension C-RASP[MOD]. C-RASP corresponds to a strict subset of $\text{UN-PLTL}_f[\overleftarrow{\#}]$, which we show in Appendix A.4. Figure 3 shows how our results embed into the existing literature.

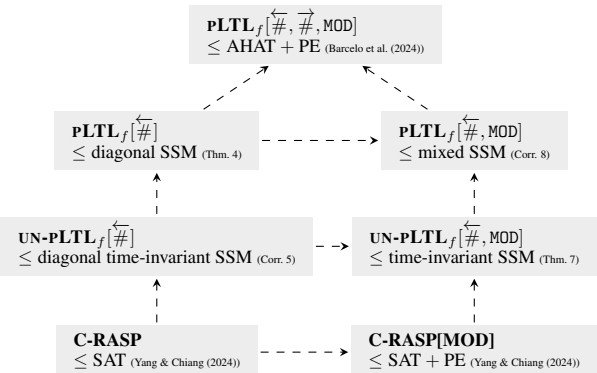

Figure 3: Comparative hierarchy under log-precision, enabling counting capabilities. The diagram positions SSM variants relative to soft-attention (SAT) (Yang & Chiang, 2024) and average hard-attention transformers (AHAT) (Barcelo et al., 2024). While SSMs strictly subsume the logic of SAT (C-RASP) due to superior local temporal processing (e.g., the *Yesterday* operator), they remain strictly less expressive than AHAT+PE. This separation arises from causality: SSMs are limited to backward-looking counting ($\overleftarrow{\#}$), whereas the global attention of AHAT allows for forward-looking counting ($\overrightarrow{\#}$).

## 7 OUTLOOK

We established lower bounds on the expressiveness of various SSM architectures by demonstrating their ability to recognise languages defined by different fragments of temporal logic. The results, as visualised in Figure 1, provide a comprehensive picture of the expressiveness hierarchy among SSM variants. While our analysis relies on explicit weight constructions, the resulting implications are robust to training methodology. Specifically, our separation results establish hard architectural limits: for instance, the monotonicity lemma implies that a diagonal fixed-precision SSM cannot be trained to recognise non-star-free languages like $(aa)^*$, regardless of the optimizer or dataset size. Furthermore, they reveal an interesting gap between the lower bounds established here and the known upper bound of $TC^0$ established by Merrill et al. (2024). The language classes that are shown here to be recognisable by various restricted SSM architectures with fixed-width arithmetic lie within $AC^0$, a proper subset of $TC^0$. This suggests two possibilities: either the $TC^0$ upper bound can be tightened to $AC^0$ for these SSM architectures, or these SSM can actually recognise languages outside of $AC^0$, such as parity. The first possibility seems more plausible, as we have not identified any mechanism in these SSM architectures that would enable counting modulo some constant—a capability required for recognising parity and other languages outside of $AC^0$. Sarrof et al. (2024) even proved that diagonal SSM cannot express parity, but only for a specific choice of non-linear layer output functions. If this is indeed the case, it would align SSM with unique hard-attention transformers which have also been shown to be limited to $AC^0$, cf. Hao et al. (2022).

Moreover, investigating potential expressiveness hierarchies based on formula nesting depth could provide finer-grained complexity classifications. Similar expressiveness hierarchies have been proved for unique hard-attention, cf. Yang et al. (2024a), and soft-attention transformers, cf. Yang et al. (2025).

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

## A APPENDIX

### A.1 EXPRESSIVE POWER OF PLTL$_f$ AND ITS EXTENSIONS/FRAGMENTS

Kamp's theorem Kamp (1968), in conjunction with results from De Giacomo & Vardi (2013); De Giacomo et al. (2021), establishes that PLTL$_f$ is equally expressive to first-order logic over words FO[$<$], which in turn corresponds precisely to the class of star-free expressions. Similarly, the logic PLTL$_f$[MOD] is expressively equivalent to first-order logic with modular predicates FO[$<$, MOD], characterising exactly the regular languages contained in AC$^0$ Straubing (1994). The unary fragment UN-PLTL$_f$ constitutes a subset of languages definable in the two-variable fragment of first-order logic FO$^2$[$<$] Etessami et al. (2002). While UN-PLTL$_f$ is restricted to the unary temporal operators *yesterday* and *previously*, FO$^2$[$<$] fully characterises languages definable in LTL$_f$ using the unary operators *yesterday*, *previously*, *next*, and *sometimes in the future*. One can show that UN-PLTL$_f$ must be a proper subset of FO$^2$[$<$] as the standard construction for separating temporal formulas into past and future components Fisher et al. (2005) relies on the *since* operator. This relationship extends to the modular predicate extensions, with UN-PLTL$_f$[MOD] forming a proper subset of FO$^2$[$<$, MOD].

Regarding the counting extension to $\text{PLTL}_f$, we observe that restricting to counting subformulas with exclusively positive weights does not enhance expressiveness. However, incorporating negative weights enables the representation of non-regular languages that may lie outside $\text{AC}^0$. For instance, consider the language $\{w \in \{a, b\}^* \mid |w|_a = |w|_b\}$ consisting of strings with equal occurrences of $a$ and $b$. This language falls outside $\text{AC}^0$ Furst et al. (1984), yet can be concisely expressed even in $\text{UN-PLTL}_f[\overleftarrow{\#}]$ with the straightforward formula $\overleftarrow{\#} a - \overleftarrow{\#} b = 0$.

## A.2 PROOFS OF SECTION 4

FEEDFORWARD NEURAL NETWORKS. An *(FNN-)node* is a function $v \colon \mathbb{R}^k \to \mathbb{R}$ with $v(\boldsymbol{x}) = relu(\sum_{i=1}^{k} c_i x_i + b)$, where $k$ is the *input dimension*, the $c_i \in \mathbb{R}$ are called *weights*, $b \in \mathbb{R}$ is the *bias* and $relu \colon \mathbb{R} \to \mathbb{R}$ with $relu(x) = \max(0, x)$ is the *activation function* of $v$. An *(FNN-)layer* $l$ is a tuple of some $n$ nodes $(v_1, \dots, v_n)$ where each node has the same input dimension $m$. It computes the function $l \colon \mathbb{R}^m \to \mathbb{R}^n$ via $l(\boldsymbol{x}) = (v_1(\boldsymbol{x}), \dots, v_n(\boldsymbol{x}))$. We call $m$ the *input* and $n$ the *output dimension* of $l$. A *Feedforward Neural Network (FNN)* $N$ consists of $k$ layers $l_1, \dots, l_k$, where $l_1$ has input dimension $m$, the output dimension of $l_i$ is equal to the input dimension of $l_{i+1}$ for $i < k$ and the output dimension of $l_k$ is $n$. The FNN $N$ computes a function from $\mathbb{R}^m$ to $\mathbb{R}^n$ by $N(\boldsymbol{x}) = l_k(l_{k-1}(\dots l_1(\boldsymbol{x}) \dots))$.

For lower bound constructions we sometimes need to make use of FNN checking specific properties, for instance: is some vector entry equal to a specific value? For easier notation we will define FNN *gadgets* for specific properties we will need later.

**Lemma 9.** *Let* $n, n_1, \cdots, n_k, m, b \in \mathbb{Z}$. *There are FNN*

- $N_{\sim b}$ *s.t.* $N_{\sim b}(n) = 1$ *if* $n \sim b$ *and* $N_{\sim b}(n) = 0$ *otherwise, for all* $\sim \in \{<, \leq, =, \geq, >\}$.

- $N_{\min(0,x)}$ *s.t.* $N_{\min(0,x)}(n) = \min(0, x)$.

*Proof.* Let $N_{=b}$ be the FNN computing $N_{=b}(x) = relu\left(relu(x - (b-1)) - 2 \cdot relu(x - b)\right)$, $N_{\leq b}$ be the FNN computing $N_{\leq b}(x) = relu\left(relu(b + 1 - x) - relu(b - x)\right)$, $N_{\geq b}$ be the FNN computing $N_{\geq b}(x) = relu\left(relu(x - (b-1)) - relu(x - b)\right)$ and $N_{\min(1,x)}$ be the FNN computing $N_{\min(1,x)}(x) = relu(x) - relu(-x) - relu(x - 1)$. Since we only work over integers, we can rewrite $x < b$ to $x \leq b - 1$ and $x > b$ to $x \geq b + 1$. It is straightforward to see that the given FNN compute the required functions. $\square$

We will sometimes need to combine FNN. Let $N_1$ and $N_2$ be FNN computing functions $\mathbb{R}^{m_i} \to \mathbb{R}^{n_i}$ for $i \in \{1, 2\}$. The composition of $N_1$ and $N_2$ is defined as follows. When $n_2 = m_1$, the FNN $N_1 \circ N_2$ computes the function $\mathbb{R}^{m_2} \to \mathbb{R}^{n_1}$ with $N_1 \circ N_2(\boldsymbol{x}) = N_1(N_2(\boldsymbol{x}))$. Syntactically, composition simply concatenates the output layer of $N_2$ and the input layer of $N_1$ setting all weights to 1.

To simplify notation, we define the gate and increment functions of diagonal SSM layers using linear projections. For fixed matrices $A, B$, let $gate(\boldsymbol{x}_t) = \text{diag}(A\boldsymbol{x}_t)$ and $inc(\boldsymbol{x}_t) = B\boldsymbol{x}_t$. This reduces the diagonal SSM layer recurrence to

$$\boldsymbol{h}_t = (A\boldsymbol{x}_t) \circ \boldsymbol{h}_{t-1} + B\boldsymbol{x}_t, \tag{1}$$

where $\circ$ denotes element-wise multiplication. We leverage the sequential decomposition of $\text{PLTL}_f$ formulas to construct an SSM evaluating subformulas layer-wise according to increasing nesting depth.

**Definition 2** (Sequential Decomposition). Let $\varphi$ be an $\text{LTL}_f$ formula with $\text{nd}(\varphi) = n$. The *sequential decomposition* of $\varphi$ is the unique sequence of sets $M_0, \dots, M_n$ with $M_i = \{\psi \in \text{Sub}(\varphi) \mid \text{nd}(\psi) = i\}$.

First, we construct the translation for each temporal operator individually, then provide the complete construction by induction over nesting depth. Given a $\text{PLTL}_f[\overleftarrow{\#}]$ formula $\varphi$ over propositions $\mathcal{P} = \{a_1, \dots, a_k\}$, we fix an injective function $\iota : \mathcal{P} \cup \text{Sub}(\varphi) \cup \{1\} \to \{1, \dots, |\mathcal{P}| + |\varphi| + 1\}$, assigning each proposition $p = a_i$ to index $i$. The assignment for remaining subformulas is arbitrary. This ordering assigns each subformula a fixed index within vectors computed by the resulting SSM,

which has dimension $d = |\mathcal{P}| + |\varphi| + 1$. The final dimension in each vector is set to the constant 1. For ease of defining linear projections, we introduce the *copy matrix* $C^{(i \leftarrow j)} \in \mathbb{R}^{d \times d}$, defined by $C^{(i \leftarrow j)} = \boldsymbol{e}_i \boldsymbol{e}_j^T$, for indices $1 \leq i, j \leq d$. Multiplying a vector $\boldsymbol{x} \in \mathbb{R}^d$ by $C^{(i \leftarrow j)}$ yields a vector whose $j$-th component is $x_i$, with zeros elsewhere.

Theorem 1 and Theorem 4 are direct consequences of the following more general result. Which we will prove at the end of this section.

**Theorem 10.** *Let $\varphi$ be a $\mathrm{PLTL}_f[\overleftarrow{\#}]$ formula over propositions $\mathcal{P}$. Then there exists an SSM $\mathcal{S}_\varphi$ over alphabet $\Sigma = 2^{\mathcal{P}}$ of dimension $d = |\mathcal{P}| + |\varphi| + 1$ with exactly $nd(\varphi)$ layers, satisfying the following: For every word $\sigma \in \Sigma^*$ with $|\sigma| = n$, if $\boldsymbol{y}_1 \cdots \boldsymbol{y}_n$ is the output produced by $\mathcal{S}_\varphi$ after the final layer on input $\sigma$, then $(\boldsymbol{y}_t)_{\iota(\psi)} = 1$ if and only if $\sigma, t \models \psi$, and $(\boldsymbol{y}_t)_{\iota(\psi)} = 0$ otherwise, for all positions $1 \leq t \leq n$ and subformulas $\psi \in Sub(\varphi)$.*

To prove this theorem, we first require some intermediate results. Specifically, evaluating formulas involving the *yesterday*-operator necessitates a suitable encoding of prior-step information into the current hidden state via the gate mechanism. This encoding was originally introduced by Sarrof et al. (2024); we provide a self-contained proof adapted to our context.

**Lemma 11** (Sarrof et al. (2024)). *Let $\sigma$ be a binary sequence $\sigma = a_1 a_2 \cdots a_n \in \{0, 1\}^*$. For every $1 \leq t \leq n$ we define the number $h_t \in \mathbb{R}$ recursively by $h_0 = 0$ and $h_t = \frac{1}{4} h_{t-1} + a_t$. There exists an FNN $N_{prev}$ which outputs the value $a_{t-1}$ at position $t$ for all $t \geq 2$, and outputs 0 at position 1.*

*Proof.* The hidden state evolves according to $h_t = \frac{1}{4} h_{t-1} + a_t$ for $t \geq 1$. Expanding this recurrence, we obtain $h_t = a_t + \frac{1}{4} a_{t-1} + \frac{1}{16} a_{t-2} + \frac{1}{64} a_{t-3} + \cdots$. The key insight is that when $a_i \in \{0, 1\}$, the value $h_t$ has the binary representation $a_t . 0 a_{t-1} 0 a_{t-2} 0 \cdots a_{1_2}$. This encoding places the current input $a_t$ in the integer part, while previous inputs $a_{t-1}, a_{t-2}, \ldots$ occupy alternating binary digits in the fractional part, separated by zeros.

To understand why this leads to disjoint intervals, observe that the contribution of all bits from position $t - 2$ and earlier is bounded: the infinite series $\frac{1}{16} a_{t-2} + \frac{1}{64} a_{t-3} + \cdots$ can be at most $\frac{1}{16} + \frac{1}{64} + \frac{1}{256} + \cdots = \frac{1/16}{1 - 1/4} = \frac{1}{12}$ when all $a_i = 1$, and at least 0 when all $a_i = 0$. Therefore, the contribution of the "tail" is bounded by the interval $[0, \frac{1}{12}]$.

Given this bound, we can analyze the four cases based on $(a_t, a_{t-1})$:

- For $(a_t, a_{t-1}) = (0, 0)$: We have $h_t = 0 + 0 + \text{tail}$, so $h_t \in [0, \frac{1}{12}]$.

- For $(a_t, a_{t-1}) = (0, 1)$: We have $h_t = 0 + \frac{1}{4} + \text{tail}$, so $h_t \in [\frac{1}{4}, \frac{1}{4} + \frac{1}{12}] = [\frac{1}{4}, \frac{1}{3}]$.

- For $(a_t, a_{t-1}) = (1, 0)$: We have $h_t = 1 + 0 + \text{tail}$, so $h_t \in [1, 1 + \frac{1}{12}] = [1, \frac{13}{12}]$.

- For $(a_t, a_{t-1}) = (1, 1)$: We have $h_t = 1 + \frac{1}{4} + \text{tail}$, so $h_t \in [\frac{5}{4}, \frac{5}{4} + \frac{1}{12}] = [\frac{5}{4}, \frac{4}{3}]$.

These intervals are clearly disjoint. For practical implementation under finite precision arithmetic, we can use slightly tighter intervals with exact binary representations. Since all boundaries can be represented exactly with at most 3 bits after the decimal point, we can safely use: $(0, 0) \mapsto [0, \frac{1}{8}]$, $(0, 1) \mapsto [\frac{1}{4}, \frac{1}{2}]$, $(1, 0) \mapsto [1, \frac{9}{8}]$, and $(1, 1) \mapsto [\frac{5}{4}, \frac{3}{2}]$. These refined intervals maintain the non-overlapping property while providing additional safety margins against rounding errors.

$N_{\text{prev}}$ extracts $a_{t-1}$ by mapping the intervals containing $(a_t, a_{t-1}) \in \{(0, 0), (1, 0)\}$ to output 0, and the intervals containing $(a_t, a_{t-1}) \in \{(0, 1), (1, 1)\}$ to output 1. This piecewise-linear function can be exactly represented by an FNN with ReLU activations using standard techniques for implementing step functions (Arora et al. (2018)). For $t = 1$, we have $h_1 = a_1$, and $N_{\text{prev}}(a_1) = 0$ by construction. For $t \geq 2$, the interval containing $h_t$ uniquely determines $a_{t-1}$, which is correctly extracted by $N_{\text{prev}}$. The factor $\frac{1}{4}$ (rather than $\frac{1}{2}$) is crucial as it provides sufficient spacing between encoded bits to prevent numerical errors under finite precision arithmetic. $\square$

Having shown how we can recover information from previous positions in order to evaluate *yesterday* formulas, we will now show how to construct layers evaluating $\mathrm{PLTL}_f[\overleftarrow{\#}]$ formulas in general.

**Lemma 12.** *Let $\sigma = \sigma_1 \cdots \sigma_n \in \Sigma^*$ be a word, $\varphi_1, \ldots, \varphi_k$ be $\mathrm{PLTL}_f[\overleftarrow{\#}]$ formulas and $\boldsymbol{x}_1 \cdots \boldsymbol{x}_n$ be a sequence with the property $(\boldsymbol{x}_t)_{\iota(\varphi_i)} = 1 \iff \sigma, t \models \varphi_i$ for all $1 \leq t \leq n$ and $1 \leq i \leq k$. Then there are diagonal gated SSM layers for all $\psi \in \{\neg\varphi_1, \varphi_1 \wedge \varphi_2, \varphi_1 \mathrm{S} \varphi_2, \mathrm{Y}\varphi_1, \mathrm{P}\varphi_1, \sum_j a_j \cdot \overleftarrow{\#}\varphi_j \sim c\}$ such that for the layer output $\boldsymbol{z}_1 \cdots \boldsymbol{z}_n$ we have $(\boldsymbol{z}_t)_{\iota(\psi)} = 1 \iff \sigma, t \models \psi$ and $(\boldsymbol{z}_t)_j = (\boldsymbol{x}_t)_j$ for all $j \neq \iota(\psi)$ and $1 \leq t \leq n$.*

*Proof.* We will show how to construct the layers for each formula individually. For all layers we set $\boldsymbol{h}_0 = \boldsymbol{0}$. When defining an FNN which only affects a single dimension in the input, we always assume that all other inputs remain unchanged.

**Case 1:** $\neg\varphi_1$**.** Notice that in order to evaluate the negation, we do not need to transfer any information from previous positions via the gate. The formula can be evaluated statically at each position using the increment function. We define

$$A = 0 \text{ and } B = I + C^{(\iota(\neg\varphi) \leftarrow \iota(1))} - C^{(\iota(\neg\varphi_1) \leftarrow \iota(\varphi_i))} .$$

After computation of the linear recurrence as defined in Eq. 1 we get a sequence $\boldsymbol{h}_1 \cdots \boldsymbol{h}_n$ with $(\boldsymbol{h}_t)_{\iota(\neg\varphi_1)} = 1 - (\boldsymbol{x}_t)_{\iota(\varphi_1)}$. As this already satisfies the claim, we can set the non-linear activation $\phi$ of this layer to be the identity $\phi(\boldsymbol{x}, \boldsymbol{h}) = \boldsymbol{h}$.

**Case 2:** $\varphi_1 \wedge \varphi_2$**.** Let

$$A = 0 \text{ and } B = I + C^{(\iota(\varphi_1 \wedge \varphi_2) \leftarrow \iota(\varphi_1))} + C^{(\iota(\varphi_1 \wedge \varphi_2) \leftarrow \iota(\varphi_2))} - C^{(\iota(\varphi_1 \wedge \varphi_2) \leftarrow \iota(1))}$$

After computation of the linear recurrence as defined in Eq. 1 we get a sequence $\boldsymbol{h}_1 \cdots \boldsymbol{h}_n$ with $(\boldsymbol{h}_t)_{\iota(\varphi_1 \wedge \varphi_2)} = (\boldsymbol{x}_t)_{\iota(\varphi_1)} + (\boldsymbol{x}_t)_{\iota(\varphi_2)} - 1 \in \{-1, 0, 1\}$. The non-linear layer output $\phi$ now maps this sequence to $\boldsymbol{z}_1 \cdots \boldsymbol{z}_n$, such that $(\boldsymbol{z}_t)_{\iota(\varphi_1 \wedge \varphi_2)} = N_{\geq 1}((\boldsymbol{h}_t)_{\iota(\varphi_1 \wedge \varphi_2)})$. This yields the wanted property.

**Case 3:** $\mathrm{P}\varphi_1$**.** In this case we need to use a gate in order to transfer information about previous positions at which $\varphi_1$ is satisfied at, to the current position.

$$A = C^{\iota(\mathrm{P}\varphi_1) \leftarrow \iota(1)} \text{ and } B = I + C^{(\iota(\mathrm{P}\varphi_1) \leftarrow \iota(\varphi_1))}$$

Notice that the gate is actually time-invariant and diagonal, as the transfer does not depend on the current input. The gate just propagates the previous value, hence

$$(\boldsymbol{h}_t)_{\iota(\mathrm{P}\varphi_1)} = (\boldsymbol{h}_{t-1})_{\iota(\mathrm{P}\varphi_1)} + (\boldsymbol{x}_t)_{\iota(\varphi_1)} = \sum_{i=1}^t (\boldsymbol{x}_i)_{\iota(\varphi_1)} .$$

Therefore, $(\boldsymbol{h}_{t-1})_{\iota(\mathrm{P}\varphi_1)}$ holds the number of times $\mathrm{P}\varphi_1$ was true at previous positions. The non-linear layer output $\phi$ now maps this sequence to $\boldsymbol{z}_1 \cdots \boldsymbol{z}_n$, such that $(\boldsymbol{z}_t)_{\iota(\mathrm{P}\varphi_1)} = N_{\geq 1}((\boldsymbol{h}_t)_{\iota(\mathrm{P}\varphi_1)})$.

**Case 4:** $\sum_j a_j \cdot \overleftarrow{\#}\varphi_j \sim c$**.** For a more compact notation we denote the formula by $\psi$. This case is basically an extension of the *previously* case. Instead of only testing if the number of positions at which a formula held is at least one, we compute a weighted sum on the number of occurences and compare it to a specific value.

$$A = C^{(\iota(\psi) \leftarrow \iota(1))} \text{ and } B = I + \sum_{j=1}^k a_j \cdot C^{(\iota(\psi) \leftarrow \iota(\varphi_j))}$$

As in the previous case this gate is also time-invariant and diagonal as it only propagates the value of a specific dimension independently of $\boldsymbol{x}_t$. We get

$$(\boldsymbol{h}_t)_{\iota(\psi)} = (\boldsymbol{h}_{t-1})_{\iota(\psi)} + \sum_{j=1}^k a_j \cdot (\boldsymbol{x}_t)_{\iota(\varphi_j)} = \sum_{i=1}^t \sum_{j=1}^k a_j \cdot (\boldsymbol{x}_i)_{\iota(\varphi_j)} = \sum_{j=1}^k a_j \cdot \underbrace{\sum_{i=1}^t (\boldsymbol{x}_i)_{\iota(\varphi_j)}}_{\overleftarrow{\#}\varphi_j} .$$

The non-linear layer output $\phi$ now maps this sequence to $\boldsymbol{z}_1 \cdots \boldsymbol{z}_n$, such that $(\boldsymbol{z}_t)_{\iota(\psi)} = N_{\sim c}((\boldsymbol{h}_t)_{\iota(\psi)})$.

**Case 5:** $\varphi_1 \, \mathtt{S} \, \varphi_2$. We use the temporal unfolding of the *since*-operator: $\varphi_1 \, \mathtt{S} \, \varphi_2 \equiv (\varphi_1 \wedge \mathtt{Y}(\varphi_1 \, \mathtt{S} \, \varphi_2)) \vee \varphi_2$. At closer look one can see that the unfolding has a similar structure as the linear recurrence in SSM layers. The current evaluation of $\varphi_1$ gets multiplied with the previous evaluation of $\varphi_1 \, \mathtt{S} \, \varphi_2$ and additionally the evaluation of $\varphi_2$ is added in order to restart the recurrence in case of a new occurence of $\varphi_2$. We define

$$A = C^{(\iota(\varphi_1 \mathtt{S} \varphi_2) \leftarrow \iota(\varphi_1))} \text{ and } B = I + C^{(\iota(\varphi_1 \mathtt{S} \varphi_2) \leftarrow \iota(\varphi_2))}$$

This leads to $(\boldsymbol{h}_t)_{\iota(\varphi_1 \mathtt{S} \varphi_2)} = (\boldsymbol{x}_t)_{\iota(\varphi_1)} \cdot (\boldsymbol{h}_{t-1})_{\iota(\varphi_1 \mathtt{S} \varphi_2)} + (\boldsymbol{x}_t)_{\iota(\varphi_2)}$. We see that in this case the gate is not time-invariant as it actually depends on the value of $(\boldsymbol{x}_t)_{\iota(\varphi_1)}$. Obviously $\varphi_1 \, \mathtt{S} \, \varphi_2$ is satisfied at some position $t$ if $(\boldsymbol{h}_t)_{\iota(\varphi_1 \mathtt{S} \varphi_2)} > 0$. The non-linear layer output $\phi$ now maps this sequence to $\boldsymbol{z}_1 \cdots \boldsymbol{z}_n$, such that $(\boldsymbol{z}_t)_{\iota(\varphi_1 \mathtt{S} \varphi_2)} = N_{\min(1,x)}((\boldsymbol{h}_t)_{\iota(\varphi_1 \mathtt{S} \varphi_2)})$.

**Case 6:** $\mathtt{Y} \, \varphi_1$. We use the encoding trick from Lemma 11. Let

$$A = \frac{1}{4} C^{(\iota(\mathtt{Y} \varphi_1) \leftarrow \iota(1))} \text{ and } B = I + C^{(\iota(\mathtt{Y} \varphi_1) \leftarrow \iota(\varphi_1))}$$

We get $(\boldsymbol{h}_t)_{\iota(\mathtt{Y} \varphi_1)} = \frac{1}{4}(\boldsymbol{h}_{t-1})_{\iota(\mathtt{Y} \varphi_1)} + (\boldsymbol{x}_t)_{\iota(\varphi_1)}$. Let $N_{prev}$ be the FNN defined in Lemma 11. The non-linear layer output $\phi$ now maps this sequence to $\boldsymbol{z}_1 \cdots \boldsymbol{z}_n$, such that $(\boldsymbol{z}_t)_{\iota(\mathtt{Y} \varphi_1)} = N_{\text{prev}}((\boldsymbol{h}_t)_{\iota(\mathtt{Y} \varphi_1)})$. This recovers the previous value of $\mathtt{Y} \, \varphi$ from position $t-1$ and yields the wanted property. $\qquad \square$

We are now ready to prove Theorem 10. The proof shows that we do not need a single layer for each subformula of $\varphi$, but it is possible to evaluate all subformulas of the same nesting-depth simultaneously. We therefore need a decomposition of every $\mathrm{PLTL}_f$ formula, by its nesting-depth.

*Proof of Thm. 10.* Let $\varphi$ be a $\mathrm{PLTL}_f[\overleftarrow{\#}]$ formula with $\mathrm{nd}(\varphi) = k$ and $M_0, \cdots, M_k$ be its sequential decomposition. We show the claim via an induction over the sequential decomposition and nesting depth $k$ of $\varphi$.

**Case $k = 0$:** $M_0$ contains only atomic formulas. Let $emb : \Sigma \to \mathbb{R}^d$ with $emb(\sigma) = \boldsymbol{x}_i$ with $(\boldsymbol{x}_i)_{\iota(1)} = 1$ and $(\boldsymbol{x}_i)_{\iota(p)} = 1$ if $p \in \sigma$ and otherwise $0$ for all $p \in \mathcal{P}$. All other dimensions are set to zero. Let $\sigma \in \Sigma^*$ be some word and $\boldsymbol{x}_1 \cdots \boldsymbol{x}_n = emb(\sigma)$, then obviously $(\boldsymbol{x}_t)_{\iota(p)} = 1 \iff \sigma, t \models p$ for all $p \in \mathcal{P}$ and $1 \leq t \leq |\sigma|$.

**Case $k > 0$:** Using the induction hypothesis we assume that the claim holds for all formulas in $M_{k-1}$. Hence, we can apply Lemma 12 to each $\psi \in M_k$. Let $(A_\psi, B_\psi)$ the *gate* and *inc* tuples gained from Lemma 12 for each $\psi \in M_k$. We now define layer $l_k$ with $A = \sum_{\psi \in M_k} A_\psi$ and $B = \sum_{\psi \in M_k} B_\psi$. Recall that Lemma 12 shows that each $A_\psi$ and $B_\psi$ only affects the dimension of $\psi$, leaving all other dimensions unchanged. This enables us to simply accumulate the effect of all gates and *inc*, resulting in a layer which simultaneously evaluates all $\psi \in M_k$. The same holds for the non-linear outputs. We argued that for each $\psi$, the layer output $\phi_\psi$ only affects the dimension of $\psi$. Hence, we can compose all FNN without affecting the wanted output. Therefore, let $\phi = \bigcirc_{\psi \in M_k} \phi_\psi$, where $\bigcirc$ denotes the repeated composition of FNNs. Using these arguments together with Lemma 12 we get that for every word $\sigma \in \Sigma^*$, the output $\boldsymbol{z}_1^l \cdots \boldsymbol{z}_{|\sigma|}^l$ of $S_\varphi$ after layer $l_n$ has the property $(\boldsymbol{z}_t^l)_{\iota(\psi)} = 1 \iff w, t \models \psi$ for all $\psi \in M_n$ and $1 \leq t \leq |\sigma|$. $\qquad \square$

*Proof of Thm. 1.* Let $\sigma = \sigma_1 \cdots \sigma_n \in \Sigma^*$ be a word and $\boldsymbol{z}_1 \cdots \boldsymbol{z}_n$ be the vector sequence produced after the last layer of $S_\varphi$ as constructed in Theorem 10. We have $(\boldsymbol{z}_t)_{\iota(\psi)} = 1 \iff w, t \models \psi$ for all $\psi \in \mathrm{Sub}(\varphi)$ and $1 \leq t \leq n$. Therefore, $\sigma \in L(\varphi) \iff \sigma, n \models \varphi \iff (\boldsymbol{z}_n)_{\iota(\varphi)} = 1$. We define the final output FNN of $S_\varphi$ such that $y_t = N_{=1}((\boldsymbol{z}_t)_{\iota(\varphi)})$. We get $\sigma \in L(\varphi) \iff S_\varphi(\sigma) = 1$. $\qquad \square$

It is important to note that the expressiveness of SSM evaluating $\mathrm{PLTL}_f[\overleftarrow{\#}]$ formulas with counting subformulas depends crucially on the arithmetic model used for computation. When operating over fixed-width arithmetic with a constant number of bits, SSM can only evaluate counting subformulas $\sum_{j=1}^k a_j \cdot \overleftarrow{\#} \varphi_j \sim c$ in which all coefficients $a_j$ are positive (i.e., formulas in $\mathrm{PLTL}_f[\overleftarrow{\#}]^+$). This restriction arises because the accumulation can only increase when coefficients are solely positive, and

the comparison against the threshold $c$ remains stable even under bounded-precision approximations. A decision procedure simply needs to determine whether the accumulated sum exceeds, equals, or falls below the constant $c$, which can be achieved with fixed precision. In contrast, when the arithmetic precision grows logarithmically with the input length, SSM can evaluate general counting formulas in $\text{PLTL}_f[\overleftarrow{\#}]$, including those with negative coefficients $a_j$. This is the case because logarithmic precision provides enough bits to exactly count occurrences of subformulas in words of length $n$, maintaining the precise difference between positive and negative terms. For instance, recognising the language $\{a^n b^n \mid n \geq 0\}$ expressed by formula $\text{H}(a \to \neg \text{Y}\, b) \wedge (\overleftarrow{\#}\, a - \overleftarrow{\#}\, b = 0)$ requires tracking the exact difference between occurrences of $a$ and $b$, which necessitates unbounded precision as $n$ grows, since any fixed-precision arithmetic would eventually lead to overflow or rounding errors when processing sufficiently long inputs.

### A.3 PROOFS OF SECTION 5

For time-invariant SSM, the gate of each layer is a constant matrix, independent of the input. For the construction carried out here we assume, as in the previous section, that the increment function is simply computed by a linear projection. This means we can represent the recurrence of a time-invariant SSM layer as

$$\boldsymbol{h}_t = A\boldsymbol{h}_{t-1} + B\boldsymbol{x}_t \,.$$

For easier construction we show that we effectively only need to maintain one single counter, even when several modular predicates occur in a formula.

**Lemma 13.** *For every $\varphi \in \text{UN-PLTL}_f[\overleftarrow{\#}, \text{MOD}]$ using modular predicates with divisors $d_1, \ldots, d_k$ there is a formula $\varphi' \in \text{UN-PLTL}_f[\overleftarrow{\#}, \text{MOD}]$, which only uses modular predicates with the same divisor $d'$.*

*Proof.* Take the least common multiple $d' = \text{lcm}(d_1, \cdots, d_k)$. Then construct $\varphi'$ by replacing each predicate $\text{MOD}_r^{d_i}$ by $\text{MOD}_r^{d'} \vee \text{MOD}_{r+d_i}^{d'} \vee \text{MOD}_{r+2d_i}^{d'} \vee \cdots$. $\square$

We will now show that every language recognised by a $\text{UN-PLTL}_f[\overleftarrow{\#}, \text{MOD}]$ formula can also be recognised by a time-invariant SSM. We will show this by extending the proof of Theorem 10. Let $\varphi \in \text{UN-PLTL}_f[\overleftarrow{\#}, \text{MOD}]$ be a formula over modular predicates all having a single divisor $m$. The resulting SSM will have $d = |\mathcal{P}| + |\varphi| + m + 1$ dimensions. We refine the function $\iota$ to a function $\iota : P \cup \{\text{MOD}_r^m \mid 0 \leq r < m\} \cup \text{Sub}(\varphi) \cup \{1\} \to \{1, \cdots, d\}$ which has the same properties as in Section 3. The only difference is that we have additional dimensions for every possible modular predicate $\text{MOD}_r^m$. For simplicity, we ensure that the indices of modular predicates are adjacent in memory, with $\iota(\text{MOD}_r^m) = \iota(\text{MOD}_0^m) + r$ for all $0 \leq r < m$, forming a contiguous interval of indices. We are then ready to prove that time-invariant SSM can recognise all languages definable in $\text{UN-PLTL}_f[\overleftarrow{\#}, \text{MOD}]$.

**Theorem 14.** *Let $\varphi$ be a $\text{UN-PLTL}_f[\overleftarrow{\#}, \text{MOD}]$ formula over $\mathcal{P}$ using only modular predicates with divisor $m$. There is an SSM $\mathcal{S}_\varphi$ over the alphabet $\Sigma = 2^{\mathcal{P}}$ with dimension $d = |\mathcal{P}| + m + |\varphi| + 1$ and $nd(\varphi) + 1$ layers, such that for all words $\sigma \in \Sigma^*$ with $|\sigma| = n$: If $\boldsymbol{y}_1 \cdots \boldsymbol{y}_n$ is the output of $\mathcal{S}_\varphi$ after the last layer on input $\sigma$, then $(\boldsymbol{y}_t)_{\iota(\psi)} = 1$ if and only if $w, t \models \psi$ and otherwise 0, for all $1 \leq t \leq n$ and $\psi \in \text{Sub}(\varphi)$.*

*Proof.* The proof follows the exact same lines as in Theorem 10. The only difference is that we need an additional layer right after the embbeding in order to evaluate the modular predicates. Let $\sigma \in \Sigma^*$ and $\boldsymbol{x}_1 \cdots \boldsymbol{x}_n$ be the vector sequence following the embedding. We add an additional layer $l_{\text{MOD}^m}$ as described in Lemma 6. For this layer, we extend the permutation matrix $P$ from Lemma 6 to the full dimension $d$ by embedding it in a larger matrix. Specifically, we construct a $d \times d$ matrix where the cyclic permutation $P$ is applied only to the dimensions $\iota(\text{MOD}_r^m)$ for $0 \leq r < m$, which form a contiguous block of indices as established earlier. All other entries are set to zero. Let $A$ be this matrix. We set the increment $B$ to be the matrix which looks like the identity matrix, but with zeros on the diagonal for all dimensions $\iota(\text{MOD}_r^m)$ for $0 \leq r < m$. Therefore we get $(\boldsymbol{h}_t)_{\iota(\text{MOD}_r^m)} = 1 \iff t \equiv r \pmod{d}$ and $(\boldsymbol{h}_t)_i = (\boldsymbol{x}_t)_i$ for all other dimensions $i$. Therefore, all

atomic formulas are evaluated after the first layer. The remaining layers are defined as in Theorem 10. □

Theorem 7 now follows directly from the previous theorem.

It is important to note that the considerations regarding fixed-width arithmetic for time-invariant SSM mirror those discussed for diagonal SSM in Appendix A.2. When operating over fixed-width arithmetic with a constant number of bits, time-invariant SSM can only reliably evaluate counting subformulas with positive coefficients (i.e., formulas in $\text{UN-PLTL}_f[\overleftarrow{\#}, \text{MOD}]^+$), as the accumulation only increases and comparisons against thresholds remain stable under bounded-precision approximations. The ability to evaluate formulas with negative coefficients requires arithmetic precision that grows logarithmically with input length. The addition of modular predicates does not affect this fundamental limitation, as the modular counter component operates independently from the counting mechanisms. Thus, when restricted to fixed-width arithmetic, the expressiveness of time-invariant SSM seems to remain limited to $\text{UN-PLTL}_f[\text{MOD}]$, which corresponds to a fragment of $\text{FO}^2[<, \text{MOD}]$.

## A.4 Proofs of Section 6

C-RASP was first introduced by Yang & Chiang (2024) as a logic called $K_t[\#]$. The syntax of C-RASP is very similar to $\text{UN-PLTL}_f[\overleftarrow{\#}]$, with the only difference being that C-RASP has no *yesterday* and no *previously* operator. The only temporal operator is the $\overleftarrow{\#}$ operator in counting formulas.

We show that C-RASP is strictly less expressive than $\text{UN-PLTL}_f[\overleftarrow{\#}]$. The proof is based on the fact that while C-RASP can simulate the *previously*-operator, it cannot simulate *yesterday*.

**Lemma 15.** *C-RASP* $\equiv \text{UN-PLTL}_f[\overleftarrow{\#}]$ *without the yesterday-operator.*

*Proof.* The inclusion of C-RASP in $\text{UN-PLTL}_f[\overleftarrow{\#}]$ is trivial, as all operators of C-RASP are also in $\text{UN-PLTL}_f[\overleftarrow{\#}]$. For the other direction, let $\varphi$ be a $\text{UN-PLTL}_f[\overleftarrow{\#}]$ formula without the *yesterday*-operator. We only need to replace all *previously* subformulas $\text{P}\,\psi$ by the counting formula $\overleftarrow{\#}\,\psi \geq 1$. This yields a C-RASP formula which is obviously equivalent to $\varphi$. □

To see that C-RASP is strictly less expressive than $\text{UN-PLTL}_f[\overleftarrow{\#}]$, we consider the language $L = \{w \in \{a, b\}^* \mid w \text{ contains } aa\}$, which is definable by the $\text{UN-PLTL}_f[\overleftarrow{\#}]$ formula $\text{P}(a \wedge \text{Y}\,a)$, but as noticed by Yang & Chiang (2024) cannot be defined in C-RASP.

