# OpenReview forum: "On the Expressiveness of State Space Models via Temporal Logics"
_ICLR.cc/2026/Conference — ICLR 2026 Poster_

### Official Review · Reviewer_FYk9 · 2025-10-23

**Soundness:** 2
**Presentation:** 1
**Contribution:** 2
**Rating:** 2
**Confidence:** 2

**Summary:**

This paper presents a theoretical analysis of the expressive power of State Space Models (SSMs), establishing a formal hierarchy of their capabilities based on gating mechanisms (diagonal vs. time-invariant) and arithmetic precision (fixed vs. logarithmic). The core contribution is a set of lower bounds linking different SSM variants to specific fragments of temporal logic (PLTLf and its extensions), thereby characterizing which classes of formal languages each variant can recognize.

The work is purely theoretical, focusing on abstract computational expressiveness rather than practical performance or model design. It provides no empirical validation, engineering insights, or direct applications to real-world tasks. The motivation stems from a foundational interest in comparing SSMs to transformers within a formal logic framework, continuing a line of theoretical research in deep learning. While it clarifies the theoretical landscape, the paper's findings have no immediate or clear practical application for practitioners designing or training SSMs.

**Strengths:**

The paper's primary strength is its “systematic” and “rigorous” approach. It aims to establish “precise”, “formal” connections between SSM variants and both well-understood and less well-known fragments of temporal logic (PLTLf), not adequately justified.

The work establishes a hierarchy of expressiveness. It differentiates the capabilities of diagonal-gated, time-invariant, and mixed SSMs, and further shows the critical impact of arithmetic precision (fixed vs. logarithmic).

It provides constructive proof sketches (proofs are in the appendix) showing how to translate a temporal logic formula into a corresponding SSM architecture. This offers some insight into the mechanisms (e.g., gates, recurrence) that enable specific computational behaviors.

The paper provides a formal comparison between the two competing architectural paradigms. It shows how the recurrent, state-based nature of SSMs leads to a different expressive profile compared to the attention-based transformers.

**Weaknesses:**

This is a purely theoretical work, and as such it expected to be rigorous. However, many terms are not explained (e.g. $\Sigma$ at line 84, $\mathcal{P}$ at line 132, $H \varphi$, $w$ at line 184). Not clear why the different logics where introduced, they appear out of the blue and it is not clear their practical importance. Why the considered extensions of pLTLf have been introduced?

Its most significant weakness from a practical standpoint is the absence of any empirical evidence. The results are "in vitro" – they show what is possible in a formal model, but not what is practically efficient, learnable, or scalable with gradient descent on real data.

The analysis is decoupled from the practical motivations that drive SSM development (e.g., training speed, long-context handling, inference efficiency). It remains unclear how these theoretical expressiveness limits correlate with the models' performance on real-world tasks like language modeling or reasoning. A practitioner would not know if reaching for a more expressive (theoretically) SSM variant would yield any tangible benefits.

The paper focuses exclusively on language recognition, a very specific and limited task. It does not address the expressive power of SSMs for sequence-to-sequence tasks, generation, or their ability to compute functions over sequences, which are more relevant to their practical use.

The constructions rely on idealized components, such as hand-crafted weight matrices and specific non-linear functions. It does not address whether these precise configurations can be discovered through standard training procedures or if they are stable under stochastic gradient descent.

The heavy reliance on formal logic, circuit complexity, and automata theory makes the work largely inaccessible to a broad machine learning audience, limiting its immediate impact outside a specialized theoretical community.

**Questions:**

The paper introduces several extensions of PLTLf (e.g., PLTLf[#↓], PLTLf[MOD]) without a clear, upfront motivation. Could the authors explicitly justify why these specific logical fragments are the most relevant or natural for analyzing SSMs? For instance, what is the intrinsic property of SSMs that makes the pure-past, backward-looking counting operator #↓ a better fit than a future-looking one?

Several key symbols are used before being defined or are used inconsistently (e.g., Σ at line 84, P at line 132, w at line 184, H φ). For a paper aiming for precision, this hinders readability. Could the authors add a "Preliminaries" section or a notation table to ensure every symbol is formally defined upon first use?

The paper states that its results "subsume" the lower bounds of Sarrof et al. (2024). Could the authors provide a more detailed discussion, perhaps with a concrete example, of exactly what new expressiveness their construction captures that the prior work did not?

The constructive proofs show that an SSM exists which can recognize a given language. However, it relies on hand-crafted, precise weights. What is the authors' intuition or any theoretical insight on whether such configurations can be discovered via gradient-based learning, or are they likely to be in regions of the parameter space that are hard to reach?

The constructions, especially for fixed-precision arithmetic, seem to depend on exact numerical values to avoid state collisions. How robust are these constructions to the inevitable noise and approximation errors that occur during stochastic training? Would a small perturbation to the gate weights cause the SSM to fail on the languages it is supposed to recognize?

The entire analysis is framed around formal language recognition. How do the authors envision these theoretical results informing the design of SSMs for core practical tasks like next-token prediction (a function computation problem) or sequence-to-sequence translation? For example, does the inability of a fixed-precision diagonal SSM to recognize (aa)* imply a fundamental limitation on its ability to model certain syntactic or semantic structures in natural language?

The paper maps SSMs to a known hierarchy of circuit complexity and logic. However, for a practitioner, the key question is: "Which SSM variant should I use for my problem?" Could the authors speculate on what their expressiveness hierarchy implies for real-world performance? For instance, does the ability to express modular predicates (like in time-invariant SSMs) suggest an advantage for tasks requiring periodicity or hierarchical structure?

While a purely theoretical paper need not include large-scale experiments, could the authors propose a minimal, synthetic experiment that could falsify or corroborate their theoretical predictions? For example, training different SSM variants on a synthetic dataset for (aa)* or $a^n b^n$ and observing if the results align with the theoretical capabilities (e.g., diagonal SSMs failing to learn (aa)*).

The heavy reliance on temporal logic and circuit complexity can limit the paper's impact. Have the authors considered adding a more extensive, intuitive overview in the introduction that explains the "story" and the practical implications of the different logic fragments (e.g., "PLTLf corresponds to pattern X, while adding counting allows for pattern Y") before diving into the formal details? A summary figure comparing the practical capabilities (e.g., "can detect repeated patterns," "can count events") of each SSM variant would be immensely helpful.

The sentence “We use this restriction, because in contrast to transformers, where the attention mechanism is usually able to attend to all positions, SSM can only see information from previous positions.” at lines 157-159 is unclear. Please clarify its meaning and why this sentence is important; similarly, for the other sentence at lines 166-167, whose role in the paper is not known.

---

> ### Author Response · Authors · 2025-11-20
>
> We thank the reviewer for the careful reading of our paper and for the many constructive suggestions. Some of the broader concerns about scope, practical relevance, and the relation to training and experiments are similar to what other reviewers have expressed. We address these in detail in a
> separate *General Comment on scope, practical relevance, and training*. Here, we briefly summarise the main points of that general comment where they directly address your criticisms, and then respond to your more specific questions and requests for clarification.
>
>
>
> **1. Scope, practical relevance, and lack of experiments**
>
> You criticise that the paper is “purely theoretical”, “in vitro”, and decoupled from practical concerns (training, efficiency, long-context handling), and that it provides no empirical evidence.
>
> As we explain in the general comment, our goal is explicitly foundational: to place SSMs in the same formal-expressiveness landscape that has recently been developed for various transformer architectures, not to propose a new architecture or training algorithm. The style of our results directly mirrors the
> existing expressiveness literature on UHAT/AHAT and soft-attention transformers: all of those also use hand-crafted weights as *witnesses* that certain behaviours are representable, and they likewise do not analyse trainability or optimisation dynamics.
>
> We hope this makes clearer how our “in vitro” analysis is intended to inform practice: as with prior work on transformers, expressiveness results give architectural guidance (“this class of models can/cannot capture this type of pattern regardless of training”), rather than prescribing specific training recipes.
>
> In the revised version we have ...
>
> - clarified the foundational scope in the introduction and outlook. We now emphasise more clearly that the primary contribution is a *relative* comparison by using exactly the same logical formalisms as in prior transformer work.
> - spelled out the practically relevant impossibility and separation results. We stress that the strongest immediate consequences for practice are *negative* and *architecture-level*: e.g., the monotonicity lemma for diagonal fixed-precision SSMs implies that no training procedure, regardless of data
>   or optimiser, can make such a model recognise non–star-free languages like $(aa)^*$. In contrast, time-invariant SSMs with modular counters can. Similarly, the fixed- vs. log-precision distinction draws a clean line between what quantised SSMs can represent (star-free/$\textsf{AC}^0$ patterns) and what only higher-precision variants can (global counting patterns such as $a^nb^nc^n$).
>
>
>
> **2. Motivation and explanation of the logical formalisms**
>
> You note that several logics “appear out of the blue” and that their practical importance is unclear, and you ask why we introduce exactly $\textsf{pLTL}_f$,
> $\textsf{pLTL}_f[\overleftarrow{\\#}]$, and $\textsf{pLTL}_f[\texttt{MOD}]$, and why we restrict counting to a backward-looking operator.
>
> Regarding the motivation for the different logics, please read our *general comment*.
>
> Furthermore, the use of $\overleftarrow{\\#}$ (and not $\overrightarrow{\\#}$) is tied to the *causal* nature of SSMs: unlike transformers with full sequence attention, SSMs only have access to the prefix of the input at each step, so backward counting is the natural counterpart of the forward-looking counting operators used for AHAT and SAT. The previous sentence you flagged (“We use this restriction, because…”) has been rewritten to say this more clearly. In the revision we have expanded the motivation for each logic in the introduction. We also added an intuitive “story” for the logics hierarchy and describe in words which *patterns* the different logics capture.
>
>
>
> **3. Notation, missing definitions, and readability**
>
> You point out that, in the original version, some symbols were used informally or before being clearly defined (e.g. $\Sigma, \mathcal{P}, H\varphi$) and that this hurts readability.
>
> While $\mathcal{P}$ and $H \varphi$ are defined in Lines 128 and 143, we added the explanation of $\Sigma$ to the definition of SSMs. We have carefully checked the remainder of the text for similar omissions, please let us know if you notice any other specific notation that requires definition.

---

> ### Author Response · Authors · 2025-11-20
>
> **4. Relation to Sarrof et al. (2024) and what we “subsume”**
>
> You ask for a more detailed discussion of what exactly we add beyond Sarrof et al. (2024), whose lower bound on diagonal SSM we claim to subsume.
>
> We have revised the related-work paragraph to be more explicit. Concretely, Sarrof et al. show that diagonal SSM (under a specific choice of non-linear output layer) can recognise exactly the star-free languages. Our Theorem 1 recovers this result *as a special case* of a more structured characterisation. We
> *extend* beyond Sarrof et al. along three axes:
>
> - We treat time-invariant and mixed SSMs and show how their expressiveness is characterised by $\textsf{pLTL}_f[\texttt{MOD}]$ and its unary fragments.
> - We analyse log-precision SSMs and relate them to counting logics $\textsf{pLTL}_f[\overleftarrow{\\#}]$ and $\textsf{pLTL}_f[\overleftarrow{\\#},
>   \texttt{MOD}]$, capturing languages like $a^nb^nc^n$ that are outside the regular/star-free world.
> - We integrate these results into the existing logic/circuit landscape and provide a comparison with transformer architectures like UHAT/AHAT/SAT.
>
>
>
> **5. Trainability and robustness to perturbations**
>
> You raise two related concerns: (i) our constructions rely on carefully chosen weights, and it is unclear whether gradient-based training can discover these configurations; and (ii) the fixed-precision constructions may not be robust to noise or approximation errors.
>
> Concerning the impact on practice and training, our results are intentionally orthogonal to any specific training procedure. The constructions in the proofs use hand-crafted weights only as *witnesses* that certain behaviours are representable, this is standard in the expressiveness literature and mirrors the results on transformer architectures we compare against. Our results do not make any claim that these exact
> parameter settings are easy to find via SGD. What they *do* show is that, for any training procedure, the *upper* and *lower* bounds we establish are architectural constraints: for example, the monotonicity argument for diagonal fixed-precision SSMs shows that no matter how we train such a model, it cannot implement non-monotone languages like $(aa)^*$ over arbitrarily long inputs.
>
> Regarding robustness, our fixed-precision arguments rely on qualitative properties (diagonal, non-negative gates; saturation; monotone arithmetic), not on exact real-valued parameters. Small perturbations that respect these qualitative assumptions do not change the fact that the state space is finite and that
> repeated application of the same symbol eventually leads to a fixed point, hence the core non-expressiveness results remain. A formal robustness analysis under realistic training noise is an interesting direction, but is beyond the scope of this work.
>
>
>
> **6. Beyond language recognition: functions and sequence-to-sequence tasks**
>
> You ask how a focus on language recognition relates to core tasks like next-token prediction or sequence-to-sequence translation, and whether, for
> instance, the inability of diagonal fixed-precision SSMs to recognise $(aa)^*$ implies limitations for modelling certain syntactic/semantic phenomena.
>
> While considering sequence-to-sequence modelling changes the theoretical setting from simple string acceptors to so-called transducers, the general limitations we showed in the paper remain the same. If an architecture cannot even implement the *decision* version of such a pattern (as our non-expressiveness results show), then no training procedure can make it reliably *use* that pattern in a more complex task either. Nonetheless, a theoretic anaylsis of SSMs for sequence-to-sequence tasks is an interesting direction for future-work.
>
>
>
> **7. Conceptual map of capabilities vs. a “design guide”**
>
> You ask whether we can give more concrete guidance on “which SSM variant should I use?” and suggest a summary figure listing practical capabilities (e.g., “can detect repeated patterns,” “can count events”).
>
> We agree that such an overview would be helpful, but we want to stress that our goal is **not** to provide a prescriptive “design manual” for practitioners. Instead, as we also emphasise in the *general comment*, our aim is to develop a **foundational, architecture-level comparison** of sequence
> models (SSMs and transformers) in terms of the qualitative behaviours they can and cannot express, using a common logical and circuit-theoretic lens. The only direct consequence for practitioners is that our non-expressiveness results show, which SSM-variants not to use, if you want to model certain patterns.
>
> ------
>
> We are grateful for your detailed feedback, which has led to substantial improvements in the presentation and positioning of the paper. We hope that the clarified motivation for the logics, the more explicit discussion of practical implications and future experiments, and the refined comparison to prior
> work will address your concerns and justify a higher overall score.

---

> > ### Comment · Reviewer_FYk9 · 2025-11-24
> >
> > Thanks for the clarifications.

---

### Official Review · Reviewer_M8MD · 2025-10-30

**Soundness:** 4
**Presentation:** 4
**Contribution:** 4
**Rating:** 8
**Confidence:** 4

**Summary:**

The paper investigates the expressive power of state space models (SSMs), an alternative to transformers, exploiting formal methods. Various fragments and extensions of Linear Temporal Logic (LTL) over finite traces (LTL$_f$) are used to proxy the SSM expressiveness. It investigates two aspects: the SSM gating mechanism (diagonal, depending on the input, vs time-invariant, remaining constant) and the arithmetic precision (fixed-width vs log-precision). Each setting combination recognizes different languages (e.g., SSM with time-invariant gates recognizes the past fragment of LTL$_f$ with unary past operators and the MOD operator with fixed-precision, while it recognizes the extended version of the same language with a past-counting operator with log-precision). All the results are formally proved, and I appreciate this a lot. The paper also positions and discusses its results with respect to relevant transformer architectures, adding value to the overall contribution. The paper is excellent and presents its results very clearly. I will recommend acceptance.

**Strengths:**

(S1) The paper is exquisite with a clean presentation. The mathematics is there, and the authors know what they are doing.

(S2) The paper's expressiveness study is impactful as it sets new grounds for what SSMs can express or not.

(S3) The work bridges the gap between symbolic and neural AI. For me, this kind of investigation allows us to see temporal, formal logic through a new lens.

(S4) Conjecture 1 makes sense. I don't have a formal argument, but I think the provided argument is sound.

(S5) Even if the study addresses only SSMs, the contextualization of the results with respect to other transformer architectures adds value. It closes the circle elegantly.

**Weaknesses:**

(W1) I really enjoyed the mathematical framework and arguments (with all the proofs), but I think the paper could benefit from an additional figure showcasing how such results are related (e.g., which Lemma(s) lead to which Theorem(s)). I expect this to improve the navigation of the paper's results.

(W2) The layout/presentation of Figures 1, 2, and 3 should be standardized. These figures could benefit from a more descriptive caption.

(W3) Some minor inconsistencies that I've identified:

- line 84: FNN is not defined (it is, however, defined at line 104)
- lines 148--156: pLTL$_f$ with both past- and future-counting operators seems counterintuitive. Since "p" stands for past, I would have presented it differently; for example, "LTL$_f$ with both past- and future-counting operators extends LTL$_f$ $\ldots$" and then (at line 157) I would have said that "By pLTL$_f$ with past-counting operator we denote the logic in which every counting subformula has $b_j = 0$ for all $j$, a fragment of LTL$_f$ with both past- and future-counting operators." Moreover, I would rather use $\sigma$ instead of $w$ to denote the words (as in the semantics at lines 134--140).
- line 252: missing closing period
- line 353: $\ell$ should be $l$ (see line 350)
- line 631: since $N_1 : \mathbb R^{m_1} \to \mathbb R^{n_1}$ and  $N_2 : \mathbb R^{m_2} \to \mathbb R^{n_2}$, then $N_1 \circ N_2 : \mathbb R^{m_2} \to \mathbb R^{n_1}$ if $n_2 = m_1$; thus, $\mathbb R^{m_1} \to \mathbb R^{n-2}$ is wrong. In any case, I would rather use $N_2 \circ N_1 : \mathbb R^{m_1} \to \mathbb R^{n_2}$ with $n_1 = m_2$; it is just simpler to read.
- lines 635 & 638: $\text{diag}(A \mathbf x_t)$ vs $(A \mathbf x_t)$
- line 707: it should be $\varphi_1 S \varphi_2$ instead of $\varphi S \varphi_2$
- Proofs of Lemma 12 & Theorem 10: for the cases, please use either bold or emphasize, but not both.

**Questions:**

(Q1) Can you please add an additional figure to improve the accessibility to readers to navigate the results?

(Q2) I would use different aesthetics for Figures 1, 2, and 3. For example, the "dashed" versus "non-dashed" inclusions could have symbols on the edges (e.g., $\subset$). Also, Figure 1 uses $\ge$, while Figures 2 and 3 use $\subseteq$; indeed, Figure 1 says "SSM with diagonal gates $\ge$ pLTL$_f$ with past-counting operator ($\log$-precision) (Thm. 4)", while Figure 2 says "pLTL$_f$ with past-counting operator $\subseteq$ diagonal SSM (Thm. 4)". The captions could be improved as well. These inconsistencies detract from the overall readability. Can you please improve all these aspects?

---

> ### Author Response · Authors · 2025-11-20
>
> We are very grateful for your detailed and positive review, and for recommending acceptance. We especially appreciate your comments on the clarity of the mathematics and on how the work “closes the circle” with results on transformer expressiveness, this is exactly the role we intended for the paper.
>
> Below we address your weaknesses and questions:
>
> **(W1 / Q1) Global navigation of results**
>
> We agree that a higher-level “map” of the results can help readers navigate the many lemmas and theorems. In the paper we have addressed this primarily at
> the textual level in the overview section. This gives readers a linear “guide” through the technical development. If space permits after incorporating the additional changes requested by the other reviewers, we also plan to add a compact figure that visually organises the results, in line with your suggestion. We agree that such a figure would be a useful complement to the textual overview.
>
> **(W2 / Q2) Figures 1–3: layout, aesthetics, and captions**
>
> We have revised Figures 1, 2, and 3 to standardize both their visual style and their logical content. All inclusions now use consistent symbols ($\geq$), so that Figure 1 and Figures 2–3 are no longer mixed in this respect. Moreover, we expanded the captions to be more descriptive, so that the intended message of each figure is clearer.
>
> **(W3) Minor inconsistencies**
>
> Thank you for the very careful list of small issues. We have gone through them one by one and incorperated the changes in the revised version of our paper.
>
> ---
>
> Once again, many thanks for your careful reading and constructive suggestions.

---

> > ### Comment · Reviewer_M8MD · 2025-11-25
> >
> > Dear authors,
> >
> > Thank you for your rebuttal. I agree with the other reviewers that there is a lack of empirical evidence, but, to me, this should not detract from the paper's primary focus. All in all, I'm still confident in my evaluation, and I will keep it that way.

---

### Official Review · Reviewer_hFRJ · 2025-10-31

**Soundness:** 3
**Presentation:** 2
**Contribution:** 3
**Rating:** 4
**Confidence:** 3

**Summary:**

The paper studies variants of state space models (SSMs) from a theoretical perspective and seeks to establish lower bounds on their expressivity. It builds upon prior work that provided upper bounds via circuit complexity. It uses fragments and extensions of (pure-past) linear temporal logic on finite traces (pLTLf) to capture the lower bounds.

The paper begins by presenting a formal model of SSMs and the lower bounds on the expressivity are established through the explicit construction of SSM layers that encode different logical operators. The two types of SSMs considered are SSMs with diagonal and time-invariant gates. Their results establish that SSMs with diagonal gates can capture pLTLf and the time-invariant version can capture the unary operator fragment un-pLTLf, along with modular counting. Additionally, log-precision arithmetic that scales precision logarithmically with input length can allow the models to perform counting.

**Strengths:**

The work builds upon an established literature that studies the limits of expressivity of different ML architectures from a theoretical lens. The paper studies the full breadth of different SSM variants and a clear expressivity hierarchy is established. Counterexamples have also been provided where possible to show that the lower bounds are "tight" at least from the temporal logic perspective.

**Weaknesses:**

The primary weakness is that these results do not suggest any ways to improve ML techniques and are far removed from practice.

There is a lot of room for improving the presentation. The abstract conveys the main message well but the introduction is severely lacking in details and a reader unfamiliar with formal logic/complexity theory would find it hard to comprehend. Many aspects are not provided with an explanation. For example, the introduction never discusses what it means to recognize a language or what LTL fundamentally is because the general ML community would not be familiar with temporal logics or formal languages. The presentation would be greatly improved with an extended introduction that puts more effort in explaining the results intuitively. This is challenging considering the amount of formalisms required but it is necessary.

Comments about notation:
- There are a few times when a layer is referred to by an $\ell$ instead of $l$.
- FNN abbreviation is used in line 84 before the expanded form is mentioned.
- The operators tt, ff, and H would be obvious to a person familiar with formal logic but not to everyone. They need to be provided with an intuitive explanation.
- ex. is used instead of $\exists$ or just exists which is non-standard.
- First-order logic is used but never defined.
- Circuit complexity classes are also used several times but never mentioned what they mean.

**Questions:**

Please comment on the weaknesses mentioned above.

---

> ### Author Response · Authors · 2025-11-20
>
> We thank the reviewer for the careful reading and constructive feedback.
>
> **On practical relevance and “improving ML”**
>
> Regarding the concern that our results “do not suggest any ways to improve ML techniques and are far removed from practice”, we would like to refer you to our *General Comment on scope, practical relevance, and impact on training* in the rebuttal. In short, our goal is to provide a foundational expressiveness analysis of SSMs that is *explicitly complementary* to the existing transformer expressiveness literature, using the same logical and complexity-theoretic framework.
>
> **On accessibility for non-theory readers**
>
> We very much appreciate the suggestion that the paper would benefit from a more accessible introduction for readers less familiar with logic and complexity
> theory. In the revised version we have substantially reworked the introduction to provide a more detailed, self-contained overview of the logical formalisms we use and include intuitive explanations of the main complexity-theoretic notions we rely on. For a detailed explanation of the connection between the different formalisms (linear temporal logic, first order logic, circuit complexity) we would like to refer you to the appendix A.1 in which we gave a extensive overview of these connections.
>
> We hope that this addresses your concern that “the general ML community would not be familiar with these details” and makes the contribution more accessible beyond the theory subcommunity. We would be very grateful if you could take a look at the revised introduction and let us know if there remain specific points where additional clarification would be helpful.
>
> ---
>
> We have also incorporated your remaining comments and suggestions in the main text.
>
> We thank you again for your thoughtful review. We hope that the strengthened discussion of practical relevance (via the general comment) and the substantially improved exposition for non-theory readers address your main concerns and that the revised version can now be evaluated more favourably.

---

> > ### Comment · Reviewer_hFRJ · 2025-11-24
> >
> > Thanks for the response, and the updated introduction. Based on this, and the other reviews, I am happy to increase my score (I still don't find the theoretical contribution to be exciting, but it's technically strong enough to merit acceptance IMO).

---

### Official Review · Reviewer_FusE · 2025-11-03

**Soundness:** 4
**Presentation:** 4
**Contribution:** 4
**Rating:** 8
**Confidence:** 2

**Summary:**

The paper investigates the expressive power of State Space Models (SSMs) in recognizing formal languages through Propositional Linear Temporal Logic (PLTL).
Diagonal SSM (with fixed-precision) can recognize PLTL_f, which is equivalent to star-free regular languages (FO[<]), but they cannot
recognize non-star-free languages like (aa)*.
Diagonal SSM with log-precision can use a backward counting operator, so they can handle UN-PLTL_f.
Time-Invariant SSM (fixed-precision) can recognize UN-PLTL_f[MOD] by using cyclic permutation matrices to maintain modulo counters and so they can recognize languages like (aa)* which are not star-free.
Time-Invariant with log-precision also handle backward counting.

Mixed SSM
   - Fixed-precision: Recognize all regular languages in AC0 (which is PLTL_f[MOD]).
   - Log-precision: Recognize PLTL_f.

Comparison to Transformers:
- Diagonal fixed-precision SSMs are comparable to UHAT without positional encodings (FO[<]).
- Time-invariant fixed-precision SSMs are similar to UHAT with positional encodings (FO[<,MOD]).
- Mixed log-precision SSM align with AHAT in expressiveness.

**Strengths:**

- The proofs builds on top of the  temporal logic theory and circuit complexity (AC⁰/TC⁰) using established formal frameworks from transformer expressiveness research.
- Bounds are tight: Diagonal SSMs’ impossibility to recognize `(aa)*` is proven via star-free language separation.
- Gate constructions are explicit and computable.
- Language recognition capabilities are formally verified, not empirically approximated.
- Limitations (e.g., TC⁹ upper bound not yet reduced to AC⁰ for SSMs) are presented.

**Weaknesses:**

- Empirical validation of depth hierarchies suggested but not explored.

**Questions:**

Could you provide preliminary experiments demonstrating mixed SSM’s ability/failure on synthetic languages under finite precision? If not, is such validation planned for the final version?

---

> ### Author Response · Authors · 2025-11-20
>
> We thank the reviewer for the very positive and encouraging assessment of our work.
>
> Regarding the main weakness raised, namely the lack of empirical validation of the depth hierarchies and the question whether we could provide preliminary experiments on synthetic languages under finite precision, we fully agree that such experiments would be valuable to bridge our theoretical predictions with observed behaviour. As outlined in our *general comment on scope, practical relevance, and impact on training*, however, the present paper is deliberately scoped as a theoretical and foundational study on expressiveness of SSMs, positioned as a counterpart to the existing transformer expressiveness results we build upon. More explanation on experimental evaluation can be found in the general comment.

---

### Author Response · Authors · 2025-11-20
**General comment on scope, practical relevance, and impact on training**

We thank the reviewers for their detailed feedback on the scope and the practical relevance of our work.

Regarding the stated concerns that our results are “far removed from practice”, lack empirical validation, and do not clearly explain how they relate to training and model design, we'd like to point out that our goal in this paper is explicitly foundational: we aim to place SSMs within the same formal expressiveness landscape that has recently been developed for transformers, rather than to propose a new architecture or training scheme (see Strobl et al. for an extensive overview). In this sense, our contribution is intended to complement the existing empirical SSM literature, much like the expressiveness results for UHAT/AHAT and soft-attention transformers that our work builds on and extends to SSMs.

Regarding the choice of formal models and logics: the fragments and extensions of $\textsf{LTL}_f$ we use ($\textsf{pLTL}_f, \textsf{pLTL}_f[\texttt{MOD}]$ and $\textsf{pLTL}_f[\overleftarrow{\\#}]$) are not introduced ad hoc. They are precisely the logics that already underlie the transformer expressiveness results of Yang et al. (UHAT, UHAT+PE, C-RASP) and Barceló et al. (AHAT), and they are tightly connected to standard circuit and first-order characterisations ($\textsf{FO}[<], \textsf{FO}[<,\texttt{MOD}], \textsf{AC}^0$, etc.). For an detailed explanation of these connections see Appendix A.1 in our paper. By analysing SSMs in terms of *the same* logical formalisms, we can make a clean *relative* comparison: e.g., diagonal fixed-precision SSMs align with $\textsf{FO}[<]$ (star-free languages) as UHAT without positional
encodings do, whereas time-invariant SSMs with modular counters match $\textsf{FO}[<,\texttt{MOD}]$ in the same way that positional encodings lift UHAT and AHAT. This is what allows us to say, in a mathematically precise sense, which kinds of temporal/structural patterns different SSM variants can and cannot capture, and how they compare to analogous transformer architectures.

Finally, we agree that synthetic experiments would be valuable to link our theoretical predictions to observed behaviour. However, such experiments are nontrivial to design and clearly beyond the scope of this paper: training sequence models on small alphabets is known to be delicate, and it is often difficult to distinguish genuine systematic generalisation from mere pattern matching on finite test sets. Building a robust validation framework for these phenomena, both for SSMs and for transformer models themselves, is, in our view, an important direction for future work, and our expressiveness hierarchy provides a principled basis for such empirical studies.

---

P. Barcelo, A. Kozachinskiy, A. W. Lin, and V. Podolskii, ‘Logical languages accepted by transformer encoders with hard attention‘, in *The twelfth international conference on learning representations*, 2024.

L. Strobl, W. Merrill, G. Weiss, D. Chiang, and D. Angluin, ‘What Formal Languages Can Transformers Express? A Survey‘, *Transactions of the Association for Computational Linguistics*, May 2024, doi: 10.1162/tacl_a_00663.

A. Yang, D. Chiang, and D. Angluin, ‘Masked hard-attention transformers recognize exactly the star-free languages’, in *The thirty-eighth annual conference on neural information processing systems*, 2024.

---

### Author Response · Authors · 2025-12-03
**Summary of Rebuttal (authors perspective)**

In light of the unusual review process this year, we briefly summarise the main points of concern that have been raised by some reviewers, as well as our responses to them:

**Reviewer FusE (rating 8)**

FusE was very positive about the scope and technical depth of the work, but asked for clearer positioning of our contribution and about empirical validation (depth hierarchies, finite-precision experiments). In the rebuttal and revision, we clarified that the paper is intended as a foundational expressiveness study (in the spirit of transformer expressiveness work).

**Reviewer hFRJ (rating 4, raised to 6 before revert)**

hFRJ's main concerns were the perceived lack of practical relevance and the accessibility of the paper for non-theory readers, in particular missing explanations of LTL, language recognition, FO logic, circuit classes and parts of the notation. We substantially rewrote and extended the introduction and preliminaries to provide more intuitive explanations, clarified notation, and pointed to an overview subsection for navigation. Importantly, after these changes and in light of the other reviews, hFRJ explicitly stated that they were "happy to increase [their] score" and considered the paper "technically strong enough to merit acceptance", resulting in a raise from 4 -> 6 before the revert.

**Reviewer M8MD (rating 8)**

M8MD recommended acceptance but asked for clearer global navigation of the results and more consistent figures and notation. In response, we strengthened the textual "map" of the results in the overview section, standardised the main figures (symbols, inclusions, captions) and systematically removed the minor inconsistencies and notation issues pointed out in the review.

**Reviewer FYk9 (rating 2)**

FYk9 raised broad concerns about missing empirical evidence, the motivation for the different temporal logics, the timing and clarity of several definitions and notations, the relation to Sarrof et al., and how our language-recognition results inform practical SSM design, including questions of robustness and trainability. In our rebuttal and the general comment, we clarified the explicitly foundational scope of the work and how expressiveness results constrain and guide architecture design (e.g. impossibility for non–star-free languages such as $(aa)^*$), expanded the introduction to motivate each logic via intuitive pattern classes, moved and sharpened definitions and notation to their first point of use, explained more clearly how our results subsume and extend Sarrof et al. and elaborated on robustness, trainability and implications for realistic sequence tasks. We regret that, due to the incident and the resulting freeze of the discussion phase, we could not engage in further back-and-forth discussion with this reviewer, as we believe that the majority of their criticisms are now addressed by the clarifications and revisions described above.

---

In summary reviewers FusE and M8MD gave a very positive remark. Reviewer hFRJ raised their score from 4 to 6 after we had responded to their comments and improved the paper accordingly. We put significant efforts to address the issues raised by reviewer FYk9, both in our comments and in a revised version of the paper, but unfortunately we did not receive feedback from the reviewer with regards to our clarifications and improvements in the paper, let alone a changed score.

---

### Meta-Review · Area_Chair_H8bC · 2025-12-28

**Summary:**

The following paper investigates the expressiveness of state space models (SSM) through the lens of linear temporal logic (LTL) over finite traces.  LTL is an extension of Boolean logic that allows for temporal statements (in a discrete time setting), i.e., it allows making statements such as "X is true because Y was true in the past". Evaluating such statements over final traces comes down to checking whether a given LTL logic statement is true for a finite input (the finite trace).

The paper verifies whether SSMs are capable of evaluating LTL statements. The authors show that this ability of SSMs is dependent on the underlying gating mechanism in the SSM architecture. They also show that SSMs can capture counting properties and non-regular languages (i.e. language patterns that require unbounded memory for their recognition) if the precision of the underlying number representation is unbounded.

**Reviewer Concerns:**

The reviewers raise several lines of concern. One line of concern questions the usefulness of the results as the modeled SSMs are detached from how they are used in practice (FYk9), and the work does not provide any ways to improve practical architectures or training techniques (hFRJ). A second line of concern points out the lack of empirical experiments (FuSE, FYk9). Finally, the reviewers criticize the lack of accessibility of the work  for the broader ML community(hFRJ, FYk9). Beyond these, the reviewers raise a number of smaller technical and writing concerns, such as about notation and definitions (M8MD, FYk9)

**Reviewer Scores:**

The authors largely and honestly conceded the lack of experiments and the detachment from practical training techniques, highlighting that this is not the goal of the paper. They suggested seeing it more as a contribution to the "limits" of expressiveness. While this may not satisfy the reviewers' concern fully, it is in line with concurrent work, and the question of expressiveness may be of interest to the broader community independently.

The accessibility to the ML community has been another major concern. Even after incorporating some of the reviewers' suggestions, the writing remains completely backwards. Beginning (in the first sentence) with a high-level explanation of what SSMs are, a concept familiar to most ICLR attendees, the paper feels as if it was written for a computational linguistics (or at least a more specialized formal methods) venue. For example, terminology that is used in the abstract without explanation, such as linear temporal logic or non-regular languages, may not even be known by those parts of the ML community who did not train in CS. While it may be debatable whether this is grounds for rejection, it is a missed opportunity to make results more broadly accessible, which most reviewers agree to be interesting.

---

### Decision · Program_Chairs · 2026-01-26

Accept (Poster)